# Transcriptional Control of Seed Life: New Insights into the Role of the NAC Family

**DOI:** 10.3390/ijms25105369

**Published:** 2024-05-14

**Authors:** Javier Fuertes-Aguilar, Angel J. Matilla

**Affiliations:** 1Real Jardín Botánico (CSIC), Plaza de Murillo 2, 28014 Madrid, Spain; jfuertes@rjb.csic.es; 2Departamento de Biología Funcional, Universidad de Santiago de Compostela, 14971 Santiago de Compostela, Spain

**Keywords:** TF binding sites, WRKY and NAC families, endoplasmic reticulum, endosperm, phytohormones, seed dormancy and germination

## Abstract

Transcription factors (TFs) regulate gene expression by binding to specific sequences on DNA through their DNA-binding domain (DBD), a universal process. This update conveys information about the diverse roles of TFs, focusing on the NACs (NAM-ATAF-CUC), in regulating target-gene expression and influencing various aspects of plant biology. NAC TFs appeared before the emergence of land plants. The NAC family constitutes a diverse group of plant-specific TFs found in mosses, conifers, monocots, and eudicots. This update discusses the evolutionary origins of plant NAC genes/proteins from green algae to their crucial roles in plant development and stress response across various plant species. From mosses and lycophytes to various angiosperms, the number of NAC proteins increases significantly, suggesting a gradual evolution from basal streptophytic green algae. NAC TFs play a critical role in enhancing abiotic stress tolerance, with their function conserved in angiosperms. Furthermore, the modular organization of NACs, their dimeric function, and their localization within cellular compartments contribute to their functional versatility and complexity. While most NAC TFs are nuclear-localized and active, a subset is found in other cellular compartments, indicating inactive forms until specific cues trigger their translocation to the nucleus. Additionally, it highlights their involvement in endoplasmic reticulum (ER) stress-induced programmed cell death (PCD) by activating the vacuolar processing enzyme (VPE) gene. Moreover, this update provides a comprehensive overview of the diverse roles of NAC TFs in plants, including their participation in ER stress responses, leaf senescence (LS), and growth and development. Notably, NACs exhibit correlations with various phytohormones (i.e., ABA, GAs, CK, IAA, JA, and SA), and several *NAC* genes are inducible by them, influencing a broad spectrum of biological processes. The study of the spatiotemporal expression patterns provides insights into when and where specific *NAC* genes are active, shedding light on their metabolic contributions. Likewise, this review emphasizes the significance of NAC TFs in transcriptional modules, seed reserve accumulation, and regulation of seed dormancy and germination. Overall, it effectively communicates the intricate and essential functions of NAC TFs in plant biology. Finally, from an evolutionary standpoint, a phylogenetic analysis suggests that it is highly probable that the WRKY family is evolutionarily older than the NAC family.

## 1. Introduction

The regulation of gene expression is a pivotal mechanism that enables plants to systematically organize their growth and development. This intricate process is governed by two essential components: trans-acting factors and cis-acting elements [1,2]. Thus, the cis-acting element in the promoter region regulates the precise initiation of gene transcription and transcriptional efficiency through the binding of transcription factors (TFs), likely determining the core region of transcriptional activation [3,4,5]. The transcriptional regulation of gene expression is directed by the action of specific proteins known as TFs, which interact with the cis-acting elements of key target genes to modulate determined signaling pathways [6]. TFs are fundamental controllers of cellular regulation, functioning in a complex and combinatorial manner, often grouping into modules. TFs bind DNA by recognizing specific sequence motifs, known as TF binding sites (TFBS), located in the promoters of target genes, thereby differentially regulating their effectiveness [7]. Consequently, the identification of TFBS holds immense importance in unraveling gene regulation during the growth and development of a specific species. TFs play a crucial role in regulating gene expression in both animals and plants, directly influencing almost all biological processes [8]. These proteins direct temporal and spatial expressions necessary for normal development and an adequate response to physiological or environmental stimuli. For a protein to be considered a typical TF, it must possess three main properties: nuclear localization, transactivation activity, and DNA-binding activity, enabling transcriptional modification. In summary, transcriptional regulation is the most extensively studied mechanism of gene regulation [8,9].

In addition to yeast one- (Y1H) and two-hybrid (Y2H) techniques, chromatin immunoprecipitation (ChIP) is a robust methodology for evaluating in vivo interactions of proteins with specific regions of genomic DNA under physiological conditions and estimating the density of TF binding at specific loci [10,11,12]. In addition to the previously mentioned techniques, the RNA-sequencing (RNA-Seq) platform [13] is widely employed to measure the transcript levels of TFs and obtain information about genome-wide binding sites. The outcome of DNA-TF binding is the activation or repression of target gene transcription. The status of chromatin is crucial for the function of TFs. Therefore, the use of ATAC-seq, a method for assessing chromatin accessibility across the genome, is recommended. Interestingly, despite TFs representing only a small fraction of plant transcriptomes (e.g., approximately 10% in Arabidopsis), they exert control over the global gene expression and regulation of a specific organism. Approximately 2296 TFs were cataloged in the PlantTFDB database and categorized into 58 families [14]. Notable among these families are DOF (DNA binding with one finger) [15], WRKY (named for its WRKYGQK sequence at their N-terminus) [16], MYB/MYC (bHLH) [17], NAC (NAM-ATAF-CUC), bZIP (basic region/leucine zipper) [18], HSF (heat shock factors) [19], DREB (dehydration-responsive element binding) [20], and AP2-EREBP (APETALA 2/ethylene (ET) response element binding protein) [21]. Other families are classified based on DNA structural domains.

NAC TFs are present in higher plants, as well as in a few species of algae. The NAC family, discovered by Souer et al. (1996) [22], stands out as one of the largest plant TF families. Land plants evolved from an ancestral charophycean alga, inheriting developmental, biochemical, and cell biological traits [23]. The NAC family is present in both aquatic green algae and higher terrestrial plants. In other words, NACs evolved from streptophyte green algae to higher plants, coinciding with the transition from aquatic to terrestrial living [24]. The moss *Physcomitrella patens* (a bryophyte) possesses 35 genes encoding putative NAC TFs [25] and 38 WRKYs [26]. Notably, the NAC family is absent in unicellular green algae and sparsely exists in mosses, such as *P. patens*. Most NAC genes have been reported to function as positive stress response TF. The abbreviation NAC is derived from the first three reported genes of this family: (i) NO APICAL MERISTEM (NAM), the first NAC gene discovered in 1996 from *Petunia hybrida*; (ii) TRANSCRIPTION ACTIVATION FACTOR 1 and 2 (ATAF1/ATAF2) from *A. thaliana*; and (iii) CUP-SHAPED COTYLEDON 2 (CUC2) from *A. thaliana*, where the NAC domain was first reported [27,28]. CUP-SHAPED COTYLEDON 2 (CUC2) was demonstrated to exhibit sequence homology and functional similarity to NAM in *Petunia hybrida* and displayed partial functional redundancy with CUC1. These three aforementioned genes encode protein sequences with a highly conserved amino acid sequence at the N-terminus. ATAF1/2, on the other hand, was found to be involved in stress responses and senescence regulation in plants [29]. NAM, ATAF1/ATAF2, and CUC2 possess a similar DNA-binding domain. In Arabidopsis, it has been reported that the NAC binding site contains the consensus DNA sequence (CGT[GA]), to which other relatively distant NAC TFs also bind [30]. However, the consensus DNA sequence in soybean is C[AC]C[GA][TC][GA]CC [31].

This update compiles recent data on the significant role of the NAC family in crucial processes such as stress, leaf senescence (LS), and seed life, across model species (e.g., Arabidopsis) and agronomically important species (e.g., rice). Particularly noteworthy is the intervention of NAC TFs in seed dormancy, germination, and reserve accumulation (e.g., starch). Additionally, considerable attention is devoted to exploring the hormonal regulation of the spatial and temporal expression of these TFs, given their modular nature and complexity. Furthermore, to provide an evolutionary framework for such diversity, based on already published sequences, this review presents a detailed phylogenetic study of the NAC and WRKY families, revealing a possible evolutionary origin for the NAC gene family. Finally, the study synthesizes the most significant conclusions from a series of investigations conducted in the last three years.

## 2. Unraveling the Complex Molecular NAC Structure

NAC proteins are abundant in terrestrial plants, exhibiting wide distribution. Notably, NAC TFs play a role in the transition of both aquatic-to-terrestrial plants and vegetative-to-reproductive growth [4,24]. This underscores the clear involvement of NACs in plant evolution. Mosses and lycophytes, representing early-diverged land plants, are predicted to have fewer NAC proteins (≤30), indicating that the expansion of NAC proteins occurred after the evolution of vascular plants. Vascular plants, in contrast, possess over 100 copies in various angiosperm species [32]. From an evolutionary perspective, NACs transitioned from algae to land organisms and subsequently expanded throughout land plants. The number of proteins increased from 20–30 in mosses and lycophytes to over 100 copies in various higher plant species [33]. This expansion is evident even in several groups of streptophytic green algae, considered to be the sister group to land plants. It has been hypothesized that the NAC family dates back more than 400 million years [24,32]. In summary, the NAC family exhibits a gradual increase in size from basal streptophytic green algae to higher angiosperms. Additionally, the plant TF database includes a total of 19,997 NAC TFs from 150 species. Among these, 328 are distributed in rice, 280 in tobacco, 138 in *A. thaliana*, and 101 in tomato (http://planttfdb.gao-lab.org; [34]). The abundance of NAC proteins in these species underscores the vital role of NACs in plant evolution [35]. Furthermore, NAC proteins exhibit a modular organization [36]. Most NACs contain a highly conserved N-terminal DNA-binding domain (the NAC domain responsible for oligomerization, typically consisting of about 150–160 amino acid residues) and a variable C-terminal domain for regulating gene expression (a relatively divergent transcriptional regulatory domain; TR) [29,37,38]. Over 10% of the NAC TFs in Arabidopsis contain an α-helical transmembrane motif within their C-terminal domain. This domain typically functions as a transcriptional activator or repressor [39,40]. Notably, NAC18 possesses both activator and repressor domains [41]. Liang et al. (2014) conducted a comprehensive analysis of the transcriptional activator OsNAP, determining that its C-terminal region (amino acids 181–392) exhibits high transcriptional activation activity. In contrast, the N-terminal region of the NAC domain (amino acids 1–190) showed no activity [42]. TaNAC019-A1 possesses a very short C-terminal region, comprising fewer than 40 amino acids, and functions as a transcriptional repressor [43]. In contrast, *ZmNAC128* and *ZmNAC130* in maize have a long C-terminal region associated with transcriptional activation activities [44]. The multiple localizations and translocation traits of NAC TFs implicate various biological functions and regulatory mechanisms [45,46]. Notably, the C-terminal domain of AtNAC2 exhibits transcriptional activation activity, whereas its N-terminal domain does not [47]. In summary, the NAC N-terminal region harbors the repression domain, while the C-terminal region largely functions as the transactivation region [48]. These properties enable NACs to exhibit multiple regulatory patterns at the transcriptional, post-transcriptional, and translational levels, serving as both activators and repressors in response to developmental and environmental cues. Studies have shown that interactions between TFs and cofactors can alter DNA-binding affinity and convert activators into repressors [49]. Similarly, TFs are enriched with intrinsically disordered regions that collaborate in DNA binding specificity. Interestingly, these regions regulate transcription through either an ABA-dependent or ABA-independent pathway, modulating stress-related gene expression in plants [50,51].

The NAC domain can be subdivided into A, B, C, D, and E subdomains (Figure 1). Subdomain A is implicated in both homo- and heterodimerization. For instance, the detection of yellow fluorescence resulting from the interaction between ONAC127-YFPN and ONAC129-YFPC confirms the formation of a heterodimer by *ONAC127* and *ONAC129* in the nucleus [52]. Subdomains C and D, characterized by positive charge and high conservation, constitute the DNA-binding sites. Lysine residues within subdomain D play indispensable roles in nuclear shuttling [29,39]. Divergent subdomains B and E contribute to the functional diversity of NAC proteins [29,53]. As a result, the NAC domain is responsible for binding properties in the N-terminus, playing pivotal roles in plant–pathogen interaction, stress resistance [54], and various other key physiological functions. Concurrently, NACs often function in a dimeric form, with the dimerization site located within the NAC domain. Dimerization is essential for stable DNA binding [55] and is established by amino acid residues Leu14-Thr23 and Glu26-Tyr31. For instance, AtNAC2 functions as a dimer, predominantly expressed in roots and moderately in flowers [47]. Intriguingly, the first structure determined by X-ray diffraction of a NAC domain protein was the crystal structure of the NAC domain from *A. thaliana* (i.e., *ANAC019*) [36]. The crystal structure of the NAC rice domain has also been reported [56]. The NAC domain monomer consists of a twisted antiparallel β-sheet surrounded by two helices, with this β-sheet playing a crucial role in sequence-specific DNA recognition [39].

Most NAC TFs are localized in the nucleus, representing their active forms, where they bind DNA and carry out specific transcriptional roles [4,48,58,59,60,61]. However, a subset of NACs is found in other cellular compartments such as the plasma membrane (PM), cytoplasm, or endoplasmic reticulum (ER), indicating inactive forms (i.e., dormant state) (Figure 2). Membrane-bound NAC TFs can be identified by the presence of a distinctive transmembrane domain, directing their association with the ER [45,62,63,64,65] or PM [30,63,66,67,68,69]. A considerable number of NAC TFs in Arabidopsis possess transmembrane domains, anchoring them to membranes [45,64]. For instance, it has been demonstrated that the full-length ANAC060 protein, containing a transmembrane domain, associates with the nuclear membrane. Conversely, its truncated form lacking the transmembrane domain is localized in the nucleus [70]. In contrast, full-length ANAC040 and ANAC089 proteins are predominantly detected on PM or ER membranes, while their truncated forms lacking the transmembrane domain are found in the nucleus [66,67,71,72]. Notably, NAC103 does not have a predicted transmembrane domain [66]. After exposure to specific developmental or environmental cues, membrane-bound TFs undergo proteolytic processing and are transported to the nucleus, where they exert their regulatory function [73]. In other words, membrane-tethered proteins exhibit extranuclear localization, requiring detachment after specific signals reach the nucleus and regulate gene expression. A recent study focused on an attractive ER membrane-tethered NAC TF (MTTF) from oilseed rape (*Brassica napus* L.) [71]. This update highlights the mechanism of MTTF nuclear import as well as its functions in regulating plant growth and stress response. Significantly, the absence of the membrane-anchoring domain in the Cvi FSQ6/ANAC089 protein resulted in nuclear localization [74]. Notably, nuclear localization signals (NLSs) have been experimentally identified in NAC proteins. The NLSs, rich in positively charged amino acids, are predominantly located toward the N-terminal end in the majority of NAC proteins [75]. This work represents the first report on the presence of unique NLSs in both the N- and C-terminal regions of NAC TFs. To summarize, NLSs are mainly situated toward the N-terminal end in the majority of NAC proteins. Collectively, at least 8.60% of NAC TFs possess a transmembrane domain at both the N- and C-terminal ends. In most cases, the transmembrane domain is located toward the C-terminal end. Current knowledge also suggests that NAC TFs play a vital role in stress resistance in ER.

Through comprehensive analysis using relevant databases and software tools, a recent detailed study on *Dendrobium nobile* revealed the localization of DnoNACs as follows: 70 members in the nucleus, 6 in the chloroplast (*DnoNAC11*, *DnoNAC12*, *DnoNAC23*, *DnoNAC30*, *DnoNAC69*, and *DnoNAC80*), 3 in the cytoplasm (*DnoNAC26*, *DnoNAC74*, and *DnoNAC78*), 3 in the mitochondria (*DnoNAC55*, *DnoNAC76*, and *DnoNAC83*), and 3 in the peroxisome (*DnoNAC16*, *DnoNAC17*, and *DnoNAC43*) [78]. These locations align with those found for the *Passiflora edulis* NAC TF family [69]. Notably, *ONAC127* and *ONAC129* proteins were observed in the nucleus, cytoplasm, and membrane of rice seeds [52]. In conclusion, research on NAC TFs has underscored the significance of this protein family in plant biology. Moreover, the number of NAC TFs per genome, along with their structural and functional properties, tends to increase with the complexity of the organism. In essence, the evolution of NAC TFs is strongly linked to the adaptation of plant life from aquatic to terrestrial forms. Recent studies have unveiled that *B. napus*, with 410 encoded NAC TFs, possesses the highest number, while *Klebsormidium flaccidum* algae, with 3 encoded NAC TFs, has the lowest [75]. Genetic studies suggest that NAC TFs in all species underwent duplication, with no loss of any of these TFs. This observation suggests that NAC TFs evolved from common ancestors through orthology and experienced numerous duplication events during divergence and speciation, indicating paralogy.

## 3. Relevant Physiological Functions of NACs

NAC proteins constitute one of the largest plant-specific TF families, renowned for their pivotal roles in various metabolic pathways during plant growth, development, and stress processes [33,34,76]. Among their diverse functions, NAC proteins are key players in processes such as LS and programmed cell death (PCD) across several studied species, including Arabidopsis, rice, tomato, wheat, oilseed rape, and cabbage [33,42,79,80,81,82,83,84,85,86,87,88,89]. They are also involved in cell wall (CW) metabolism [33,48,90,91,92,93,94,95,96], reactive oxygen species (ROS) production [33,77,96,97,98], nutrient remobilization [44,99], xylem/phloem differentiation and formation [88,100,101,102], regulation of plant immunity [54], and hormone signal transduction (e.g., salicylic acid (SA), jasmonic acid (JA), ethylene (ET), and abscisic acid (ABA)) [33,34]. Furthermore, NAC proteins respond to signals such as ROS, contributing to plants’ resistance against pathogens [60,87,103]. Their involvement extends to seed life (to be further discussed) and the regulation of biotic and abiotic stresses, including drought, heat stress, low-O_2_ (anoxia), and cold tolerance [33,52,57,61,104,105,106,107,108]. As demonstrated in this update, NAC TFs emerge as dominant regulators in response to abiotic stresses. Surprisingly, the role of NAC TFs in regulating plant functions under non-stress conditions is less explored. Plants frequently encounter combined stresses rather than single stressors, leading to more severe disruptions. However, the underlying mechanisms of these combined stresses remain unknown. Notably, *IbNAC3* from sweet potato (*Ipomoea batatas*) has been found to confer tolerance to both single and combined salt and drought stresses in Arabidopsis [61]. Interestingly, ANAC102 is essential for tolerance to stress induced by low-O_2_ concentrations (i.e., 0.1% O_2_ stimulates *ANAC102* expression) during the seed stage but not in early adult plants. In other words, although *ANAC102* is not crucial for adult tolerance to low-O_2_, it plays a significant role in tolerating low-O_2_ levels during germination [104]. In the complicated regulatory network governing the CW, *NAC29* and *NAC31* play crucial roles by influencing downstream cellulose synthase (CESA) activity. These two TFs exert their regulatory effects by activating the *MYB61* TF. The activation of *MYB61*, in turn, serves as a crucial mechanism for controlling the synthesis of secondary CW [109].

### 3.1. NAC TFs Play Crucial Roles in Both ER Stress Responses and Leaf Senescence

The accumulation of unfolded or misfolded proteins can disrupt ER protein homeostasis, resulting in ER stress and compromising cellular function. In response, the unfolded protein response pathway is activated to mitigate ER stress. Additionally, ER stress is known to directly generate ROS signaling and modify the redox status. NAC TFs have been identified as key players in the response to ER stress. Under these conditions, the Arabidopsis *NAC103* transcriptional activator [66], along with the membrane-associated NAC089 and NAC062, are all upregulated. NAC103 forms homodimers in both yeast and plants and is induced by bZIP60 [110]. It plays a crucial role in the expression of genes related to DNA damage response, contributing to stabilization under unfolded protein stress [110,111]. The overexpression of *NAC103* leads to a significant increase in the transcription of ER stress-responsive genes [110]. Meanwhile, *AtNAC062*, a plasmalemma-associated TF, plays a vital role in promoting cell survival under ER stress conditions. NAC062 undergoes relocation from the plasmalemma to the nucleus, where it regulates the expression of ER stress-responsive genes [71,72]. On the other hand, *NAC089* relocates from the ER membrane to the nucleus, inducing programmed cell death (PCD) in response to ER stress treatment [71,72]. Recently, AtNAC091 was identified as a crucial unfolded protein response mediator, and its expression was demonstrated to be induced by ER stress. This induction is primarily dependent on the ER stress regulators bZIP60 and bZIP28 [112]. In contrast, the *Glycine max* transcriptional repressor GmNAC81 is positively regulated during LS. Notably, the GmNAC81-mediated response and senescence-induced response share similar mechanisms. As a consequence, overexpression of *GmNAC81* in the early vegetative stage accelerates LS and increases sensitivity to drought [113]. Furthermore, GmNAC30 and GmNAC81 (formerly GmNAC6) have been demonstrated to participate in ER stress [71,79,113]. Both TFs collaborate to activate the caspase-1-like vacuolar processing enzyme (VPE) gene, playing a crucial role in the stress-induced PCD response in plants [114]. Interestingly, GmNAC30 forms a heterodimeric interaction with GmNAC81 in the nucleus of plant transfected cells, suggesting the potential biological significance of this interaction. Mendes’s group has demonstrated several roles for both *GmNAC81* and *GmNAC30*: (i) both are induced by ER stress with similar kinetics; (ii) they are upregulated by the cell death-inducer cycloheximide and promote cell death when transiently expressed in soybean protoplasts; (iii) both TFs bind in vivo to common target promoters and regulate their expression in a coordinated manner; (iv) the VPE promoter contains a *GmNAC81/GmNAC30* binding site; and (v) *GmNAC81* and *GmNAC30* specifically bind to the core consensus sequence TGTGTT [115]. In summary, GmNAC30 and GmNAC81 work together to regulate VPE expression, a plant-specific executor of cell death. VPE is likely responsible for the execution of the cell death program induced by ER and osmotic stress.

The LS is a genetically and epigenetically programmed process that can be induced by internal and adverse environmental factors and marks the final stage of leaf development [42]. In *Glycine max*, approximately 40% of the *NAC* genes are upregulated in LS, and only *GmNAC030*, *GmNAC065*, and *GmNAC081*, have been functionally characterized. In Arabidopsis, several NAC TFs have been identified as specific regulators of LS. Notably, ATNAP, ANAC029, ANAC092 (ORE1), ANAC059 (ORS1), ANACO42 (JUB1), and ANACO83 (VN12) play prominent roles in the regulatory network governing LS. The overexpression of *AtNAP*, *ANACO59*, and *ANACO92* induces precocious senescence, while blocking the function of these three TFs significantly delays LS [42]. *ANACO92* (*ORE1*), identified as a positive regulator of LS, plays a crucial role in controlling the aging process in *A. thaliana*. The overexpression of *ORE1* leads to early senescence, whereas its inhibition retards LS [116,117]. Indeed, the *ore1-2* and *ore1-6* lines exhibit delayed LS. The interaction between ET and ORE1 in LS has been demonstrated in several manuscripts [118,119]. The increased expression of *EIN2* directly upregulates the expression of *ORE1* [119]. In contrast, the expression of *ORE1* is negatively regulated at the post-transcriptional level [117]. These authors have provided evidence of *ORE1* binding in vivo to the promoters of two other senescence-associated genes (*SAG29/SWEET15* and *SINA1*). Additional manuscripts have highlighted the direct binding of ORE1 to *PRR9*, *SAG29*, and *BFN1*, among other senescence-associated genes, to regulate its expression [83,118,119]. In a recent study, a mechanism modulating ore1 in *G. max* L. has been proposed [120]. This mutant demonstrates high tolerance to oxidative stress and significantly delays LS in *A. thaliana* [115]. It has been proposed that upon introducing *At-ore1* into *G. max*, *At-ore1* exhibits an inverse role in LS depending on the ABA/IAA balance. This effect is possibly mediated through an increase in *GmNAC081*- or *GmNAC065*-mediated H_2_O_2_ regulation [121]. Recently, it was found that *GmNAC81* overexpression in the early vegetative stage accelerated LS and increased sensitivity to drought. In addition, it was suggested that *GmNAC81* may negatively control drought tolerance not only via VPE activation but also through the suppression of ABA signaling, and that GmNAC81 regulated the expression of six target genes (e.g., *KIT1* and *STI* were highly repressed by *GmNAC81* overexpression) involved in LS [113]. Collectively, data from [113] propose that GmNAC81 directly modulates drought tolerance by controlling critical regulators of ABA signaling. In other words, *GmNAC81* may negatively regulate drought tolerance not only through VPE activation but also through the suppression of ABA signaling. In summary, current knowledge suggests that NAC TFs serve as positive or negative regulators of LS by either activating or repressing RBOH-mediated ROS production. For instance, *BrNAC055* has been shown to activate the in vivo transcription of *BrRbohB*, *BrRbohC*-like, *BrNYC1*, and *BrNYE1*, indicating its role in regulating LS through ROS-related mechanisms. These findings provide evidence to support that BrNAC055 works as an activator of *BrRbohB*, *BrRbohC*-like, *BrNYC1*, and *BrNYE1* transcription through direct binding to their promoters.

### 3.2. Functions of OsNAC2 and Other Members of OsNAC Family in Rice

Rice, being a staple food consumed by half of the human population, warrants comprehensive study at various molecular levels. The rice genome is predicted to contain 151 *NAC* genes. Among the transcription factors, *OsNAC2*, considered the ortholog of *AtORE1* (Arabidopsis NAC092), stands out as one of the most extensively studied in rice. *OsNAC2* plays a crucial role in modulating plant height and architecture by participating in GAs signaling pathways [122]. Additionally, it contributes to LS through ABA biosynthesis [123], enhances abiotic stress tolerance [11], is involved in PCD [124], and has recently been implicated in the release of seed dormancy and germination [11]. Overexpressing *OsNAC2* in plants enhances stress resistance to cold, drought, and salt, while transgenic rice with inhibited *OsNAC2* expression shows reduced salt tolerance [125]. Moreover, *OsNAC2* has been demonstrated to play a role in regulating root development by integrating auxin and cytokinin (CK) signaling pathways [126]. Additionally, *OsNAC45* is implicated in ABA response and contributes to salt tolerance [95]. Rice plants overexpressing *OsNAC2* exhibit increased transcription of ABA biosynthesis genes, namely *OsNCED3* and *OsZEP1*, with *OsNAC2* binding to their promoters [123]. Interestingly, *OsNAC2* is upregulated in response to low ABA levels and downregulated under high ABA levels, suggesting a feedback repression mechanism for *OsNAC2*. In other words, (i) *OsNAC2* functions as a repressor of yield, and flowering time in rice [123]. Additionally, OsNAC2 functions at the intersection of ABA and abiotic stress pathways; (ii) the role of *OsNAC2* contrasts with other reported NAC members such as *OsNAC5*, *OsNAC6*, and *OsNAC10*, where their overexpression has been shown to enhance drought stress resistance and grain yield [125]; (iii) overexpression of *OsNAC2* suppresses salt resistance in transgenic rice plants, increasing sensitivity to high salt stress. Additionally, alterations in *OsNAC2* expression impact drought sensitivity during the vegetative state; (iv) *OsNAC2* has the ability to directly bind to the promoter of LATE-EMBRYOGENESIS-ABUNDANT-3 (OsLEA3) and downregulate its expression; and (v) *OsNAC2* exhibits elevated expression in *OsEIN2* overexpressing lines, whereas its expression is downregulated in *ein2* mutants. *OsLEA3*, a well-known stress marker gene, has been shown to confer stress resistance when overexpressed [125]. *OsNAC2* expression is significantly induced by ABA and osmotic stress. Therefore, *OsNAC2* functions through the ABA-dependent pathway and may downregulate numerous ABA-responsive stress marker genes. It is plausible that *OsNAC2* regulates both abiotic stress responses and ABA-mediated responses, influencing the intersection between the ABA and abiotic stress pathways. Additionally, the root-specific overexpression of *SNAC1*, *OsNAC5*, *OsNAC6*, *OsNAC9*, and *OsNAC10* remarkably enhances drought resistance in rice plants at the reproductive stage. In particular, *OsNAC9* modifies root architecture, extending aerenchyma and stele, contributing to drought tolerance and enhanced grain yield under field conditions [127]. Similarly, OsNAC10 enhances stele, cortex, and epidermis size, improving drought tolerance and grain yield in field drought conditions [128]. Lastly, *OsNAC011* serves as a promoter of LS, as plants overexpressing *OsNAC011* exhibit a precocious senescence phenotype, while knockdown plants display reduced LS with a high accumulation of chlorophyll [119,123].

### 3.3. Additional Roles of NACs Genes in Other Eudicot Species

Recent findings related to *BnaNAC60* in *B. napus* indicate its involvement in various processes associated with LS and stress responses. *BnaNAC60* expression is induced during the progression of LS in *B. napus*, suggesting its involvement in the regulation of senescence-related processes [98]. BnaNAC60 is a membrane-tethered protein, indicating its initial association with cellular membranes. However, it undergoes translocation to the nucleus in response to both ER stress and oxidative stress treatments [98]. BnaNAC60 acts as an upstream regulator of cell death, LS, and ROS accumulation. It likely influences these processes by activating the promoter activity of genes involved in ROS generation (such as *BnaRbohD*) and ROS signaling (such as *BnaZAT12*) [77]. These findings suggest that *BnaNAC60* is a multifaceted regulator involved in coordinating responses to stress conditions and senescence in *B. napus*. The activation of genes related to ROS suggests a potential link between *BnaNAC60* and redox signaling, which is often crucial in stress responses and senescence. Recently, it was demonstrated that the CmNACP1-mRNA of *Cucumis melo* moves through the phloem, indicating its capacity for long-distance transport. and the little-characterized *AtNAC087* of *A. thaliana* is expressed in the phloem, apex, and developing flowers. In contrast to CmNACP1-mRNA, AtNAC087-mRNA is not mobile, suggesting a cell-autonomous function affecting the cells where it is expressed without being transported over long distances. Given that *CmNACP1* and *AtNAC087* are orthologs (i.e., a common evolutionary origin), this feature was used to investigate the cellular role of the *AtNAC087* gene [129]. The overexpression of *AtNAC087* leads to the accelerated LS of aerial rosettes, confirming its function as a positive regulator of LS in a tissue-dependent manner. Specifically, *AtNAC087* orchestrates post-mortem chromatin degradation specifically in roots via BFN1 nuclease, a senescence-associated type I nuclease. Interestingly, the tissue-dependent nature of its regulatory functions implies that *AtNAC087* may have distinct roles in different plant tissues, contributing to the complexity of senescence-related processes. Ectopic expression of *AtNAC087* correlates with the emergence of aerial rosettes, likely resulting from the induction of *HUA2*, a gene involved in flower induction [129]. Additionally, the gaseous hormone ET regulates various aspects of plant growth and development, with fruit ripening being the most well-known, along with responses to biotic and abiotic factors [130,131]. Numerous studies have indicated that NAC TFs play a significant regulatory role in fruit development and ripening, particularly in relation to ET signaling [132,133,134,135]. The recent identification of two NAC TFs, *PpNAC1* and *PpNAC5*, in Prunus represents a significant advance in understanding their role in fruit development and ripening. Orthologs *PpNAC1* and *PpNAC5* activate the transcription of genes associated with CW degradation, cell elongation, and ET biosynthesis. Interestingly, the authors suggest a critical regulatory role for both genes in the ripening of peach fruit-producing plants but not in fruitless ones [136]. On the other hand, Peng et al. (2023) [137] obtained these attractive results in *C. maxima* (pumpkin): (i) *CmoNCED6* is identified as the key gene for ABA synthesis, and *CmoNAC1* is most likely an important TF regulating H_2_O_2_ and ABA signaling in pumpkin rootstock under salt stress; (ii) *CmoNAC1* influences the production of H_2_O_2_ and ABA by regulating the expressions of *CsaRBOHD* and *CsaNCED3* in grafted cucumber leaves, as well as *CmoRBOHD1* and *CmoNCED6* in roots. Similar results were also obtained in tomatoes [23,47], cowpeas [35], soybeans [48], and Arabidopsis [37]; (iii) *CmoNAC1* binds to the promoters of *CmoRBOHD1*, *CmoNCED6*, *CmoAKT1;2*, and *CoHKT1;1* in response to salt stress in grafted cucumbers. In summary, *CmoNAC1* is not only a key TF positively regulating salt tolerance in grafted cucumbers but also promotes the synthesis of H_2_O_2_ and ABA signals in roots, contributing to an increased K^+^/Na^+^ ratio. This suggests a role for *CmoNAC1* in maintaining ion homeostasis under salt stress conditions [137].

### 3.4. Recent Novelties in Stress Response by NACs Genes

As evidenced throughout this review, NAC TFs play essential roles in stress responses. However, there is limited information available on stress-related NAC genes in maize, despite the identification of a total of 190 predicted NAC members. With respect to salt tolerance, recent research in *Zea mays* demonstrated the involvement of the *ZmNAC89* gene in stress caused by Na_2_CO_3_ [138]. Here are some key properties of *ZmNAC89*: (i) it possesses transcriptional activation activity and is a nuclear protein; (ii) it exhibits outstanding function against salt-alkali tolerance, with overexpression of *ZmNAC89* improving both stresses in *A. thaliana*; and (iii) its expression is induced by ABA, being upregulated under ABA stress [138]. These findings collectively highlight *ZmNAC89* as a stress-responsive TF with promising attributes for enhancing salt and alkali tolerance in plants. Previously, functions of other maize NAC genes have been identified. For instance, *ZmSNAC1*, *ZmNAC33*, and *ZmNAC55* significantly enhance drought tolerance in transgenic Arabidopsis, while *ZmNAC111* is associated with drought resistance [139,140,141]. Similarly, ectopic expression of *CmNAC1* enhances the tolerance of transgenic *A. thaliana* to cold, salt, and drought stresses [142]. The overexpression of *VvNAC17* from grape in Arabidopsis, leading to increased sensitivity to ABA and enhanced tolerance to salt, freezing, and drought, underscores the multifaceted roles that NAC TFs can play in plant responses to environmental stresses [143]. It is noteworthy that elevating *ZmNAC111* expression in transgenic *Z. mays* enhances water use efficiency at the seedling stage [144]. On the other hand, *ZmNAC33* is strongly induced by ABA and acts as a positive regulator in drought tolerance [141]. This suggests that *ZmNAC33* may be part of the ABA-mediated signaling pathway, contributing to the plant’s ability to cope with drought stress. Interestingly, several stress-related *cis*-acting elements exist in the promoter region of the *ZmNAC55* gene, which is induced by ABA, high salinity, and cold stresses. Similarly, overexpression of *ZmNAC55* in Arabidopsis led to hypersensitivity to ABA during seed germination [140]. In summary, *ZmNAC55* appears to be a stress-responsive NAC gene with a role in ABA signaling. Its induction under various stress conditions and its impact on seed germination highlight its potential importance in the plant’s adaptation to environmental challenges, particularly those related to ABA-mediated responses. The results obtained with *ZmNAC55* open the possibility of feasible investigations in the transgenic breeding of drought-tolerant crops. However, *ZmSNAC1* was strongly induced by low temperature, NaCl, drought stress, and exogenous ABA, while being downregulated by SA. Overexpression of *ZmSNAC1* in *A. thaliana* led to hypersensitivity to ABA and osmotic stress, conferring tolerance to dehydration without any pleiotropic effects [139]. Finally, it is noteworthy that overexpression of *ZmNAC84* in tobacco improved drought tolerance, while *ZmNAC88* played a crucial role in ABA-induced antioxidant defense against drought stress [145]. Given its remarkable impact, the study of the *ZmNAC84* function has recently advanced. Today, it is known that *ZmNAC84* physically interacts with *ZmCCaMK* (Ca^2+^/calmodulin-dependent protein kinase) both in vitro and in vivo. Functional analysis has revealed that *ZmNAC84* is essential for ABA-induced antioxidant defense in a *ZmCCaMK*-dependent manner. In other words, ABA-produced H_2_O_2_ first induces the expression of *ZmCCaMK* and *ZmNAC84*, then activates *ZmCCaMK*, which phosphorylates ZmNAC84 at Ser-113, thereby initiating antioxidant defense by activating downstream genes [145]. Much more recently, it was demonstrated in *Z. mays* that phosphorylated ZmNAC84 enhances drought tolerance by directly modulating the expression of *ZmSOD2*, a key player in the plant’s antioxidant defense against ROS generated during drought stress [146].

Three novel nuclear NAC genes from the important crop *Triticum aestivum*, *TaNAC2*, *TaNAC29*, and *TaNAC67*, were characterized for their enhanced multi-abiotic stress tolerance in transgenic plants. *TaNAC29* expression is stimulated by H_2_O_2_, salt, and ABA and exhibits high levels in LS, indicating its involvement in the senescence process and response to salt and drought stresses [147]. Interestingly, introducing the rice stress-responsive NAC gene into wheat resulted in a transgenic plant with significantly enhanced tolerance to multiple abiotic stresses [148]. On the other hand, transgenic lines overexpressing *TaNAC69* produce more biomass in the shoot and root under stress-inducing conditions, demonstrating increased resistance to salinity and drought stresses through enhanced scavenging of ROS [149]. Taken together, all the results discussed in Section 3, and others not included in it, demonstrate that a large number of *NAC* genes play a key role in enhancing abiotic stress tolerance, and their function is likely conserved in angiosperms. Moreover, these findings suggest that the use of transgenic plants appears to be an adequate approach for addressing stress tolerance in the near future through biotechnological methods [150]. Recently, 104 *NAC* genes were identified in *Camellia sinensis*, the tea plant [150]. Some of these genes possess one or two transmembrane domains at the C-terminus, while others have these domains at the N-terminus. The expression of most of them was induced by exogenous ABA and drought stress (i.e., PEG treatment). Interestingly, overexpression of *CsNAC28* in Arabidopsis increased sensitivity to ABA and resulted in additional upregulation of drought-responsive genes. In summary, [150] provides demonstrative data that *CsABF2* could bind to the ABRE cis-acting element in the promoter of *CsNAC28* and activate *CsNAC28* expression. Comparatively, *CsNAC28* transgenic plants, when compared to the WT, exhibited significantly reduced ROS content and less oxidative damage under drought stress. In other words, the improved antioxidant capability of *CsNAC28* transgenic plants enhanced drought resistance at the cellular level in transgenic Arabidopsis.

In gymnosperms, such as conifers, the molecular mechanisms that regulate multiple abiotic stress responses remain largely unnoticed. Recently, data have been obtained that point to a notable role of NAC TFs. Thus, it was demonstrated in *Pinus tabuliformis* under abiotic stresses that *PtNAC3* is nuclear, and the activation of *PtNAC3*—a stress-related NAC gene—is mediated by ET rather than ABA [60]. In short, PtNAC3 and AtNAC3 possess identical amino acid N-terminal NAC domains and are highly upregulated by ET, exhibiting co-expression. PtNAC3 protein may directly target PtZFP30, a zinc finger protein involved in abiotic stresses, activating it. *PtZFP30* is upregulated by ET and ACC, and is likely one of the downstream genes of PtNAC3 [60]. This recent work provides intriguing insights into the hub nodes of the abiotic stress response network in conifers. On the other hand, two NAC genes, namely *PpNAC2* and *PpNAC3* (which cluster with Arabidopsis *ATAF1* and *ATAF2*), were characterized from *Pinus pinaster* (maritime pine). Their expression was strongly and rapidly induced by MeJA and high salinity. *PpNAC2* and *PpNAC3* promoters contain *cis*-elements involved in biotic and abiotic stress. Similarly, NbbHLH2 proteins can mediate the JA induction of the *PpNAC3* promoter, suggesting the conservation of the JA signaling pathway between angiosperms and gymnosperms. In summary, the knowledge about NAC TFs in terrestrial plants leaves no doubt that their role has been strengthened throughout evolution, becoming particularly evident in the regulation of stress processes.

Finally, the development of bioinformatics and modern molecular biology techniques, coupled with the recognition that *NAC* genes play pivotal roles in various plant processes, has promoted extensive studies on the *NAC* gene family across numerous species (Table 1). This is the case with *A. thaliana* [151], rice [152], wheat [153], soybean [154], peanut [46], maize [155], potato [156], cassava [157], Chinese cabbage [158], pepper [159], melon [160], apple [161], sunflower [162], *Saccharum spontaneum* [163], Tartary buckwheat [141], *Zanthoxylum bungeanum* [164], *Asparagus officinalis* [165], *Brassica juncea* var. Tumida [166], *Liriodendrum* [4], *Hibiscus hamabo* Sieb [167], and other species (see Introduction from [168]). Likewise, accumulated studies have shown that there are currently 117 *NAC* genes in Arabidopsis [169], 163 in poplar [170], 74 in *Vitis vinifera* [171], 85 in *Liriodendrum* [4], 151 in rice [54], 110 in potato [156], 152 in tobacco [172], 152 in soybean [154], 97 in *Medicago truncatula* [173], 104 in tomato [53], 204 in Chinese cabbage [174], 152 in maize [155], 85 in *Dendrobium nobile* [78], 150 in *Helianthus annus* [46], 105 in passion fruit (*Passiflora edulis*) [69], and 123 in *Pinus tabuliformis* [60]. Similarly, the genome-wide identification of *NAC* genes has been extended to numerous plant species (see Introduction and Discussion in [175]).

## 4. Phytohormonal Requirements for NACs Function in Vegetative Organs and Leaf Senescence

As previously stated, NAC TFs play important roles in various biological processes and responses to abiotic stresses. Many of these tasks are regulated by hormones, and several *NAC* genes play critical roles in these processes [37,47,135]. This is so much so that various *NAC* genes are hormone-inducible [192,193,194]. To date, only a few *NAC* genes have been specifically shown to regulate LS. For example, *ANAC029*, *ANAC042*, *ANAC059* (*ORS1*), *ANAC083*, and *ANAC092* (*ORE1*) play significant regulatory roles in this aging process [129]. Overexpression of *AtNAP*, *ORS1*, and *ORE1* triggers acceleration of LS, and blocking the function of these TFs significantly delays LS. Therefore, it has been suggested that *AtNAP*, *ORE1*, and *ORS1* act as non-redundant positive regulators of LS in Arabidopsis [195]. The prematurely senile 1 (*ps1-D*), a dominant precocious LS mutant, was previously identified [42]. *PS1* (formerly called *OsNAP*) encodes a plant-specific NAC and is considered an adequate marker for the onset of the senescence process in rice. One notable finding is the connection between *OsNAP* and ABA. A feedback repression mechanism has been identified where OsNAP negatively regulates ABA biosynthesis. This is evidenced by a significant reduction in ABA content in *ps1-D* mutants, which are associated with *OsNAP*. This feedback loop implies that OsNAP not only participates in the regulation of LS and endosperm maturation but also influences ABA levels, showing its multifaceted role in plant physiology. Accordingly, the transcription level of *OsNAP* is directly linked to the nutrient remobilization associated with senescence [42].

### 4.1. The Relationship of ABA and ET with NAC

The RD26 gene was the first NAC identified as a regulator of both ABA and JA signaling during stress responses in Arabidopsis [103]. On the other hand, it is well-known that ABA synthesis and signaling upregulate LS, a stress process in which several NAC TFs are involved. NAC/ATAF1 directly regulates the ABA biosynthetic gene *NCED3* in *A. thaliana* [196], whereas *OsNAP* (NAC-LIKE, ACTIVATED BY AP3/PI) confers abiotic stress response exclusively through the ABA pathway in rice [182]. These findings indicate the intricate crosstalk between ABA and JA signaling pathways, with NAC TFs like *RD26*, *NAC/ATAF1*, and *OsNAP* playing roles in mediating stress responses and senescence through these pathways. The transcription of *OsNAP* increased rapidly, up to 102-fold after 12 h of ABA treatment, and sharply reduced in *aba1* and *aba2* mutants [42]. In summary, *OsNAP* appears to act in rice as a key regulator linking the ABA signaling and LS processes. No hormone other than ABA can regulate the OsNAP-mRNA level [42]. Understanding the specific regulatory interactions involving OsNAP and ABA highlights its importance in mediating stress responses and senescence in rice, particularly in response to ABA signaling. Likewise, overexpression of *OsNAP* also led not only to an increased expression of JA biosynthesis genes (i.e., *OsLOX2* and *OsAOC*) but also to endogenous JA in transgenic plants [183]. On the other hand, *OsNAC6* has been reported to be induced by exogenous JA [176,184], whereas *ANAC019* and *ANAC055* participate in the crosstalk between ABA and methyl jasmonate in plant defense response [176]. In tomato, SlNAP2 has a central role in controlling LS and fruit yield [84]. That is, both *SlNAP1*- and SlNAP2-mRNAs increased in the leaves during tomato dark-induced senescence. The *AtNAC2* expression pattern following ABA treatment was similar to the salt stress treatment. ABA and salinity treatments resulted in an approximately threefold induction of *AtNAC2* expression. However, the level of AtNAC1-mRNA was not influenced by salt stress, ABA, or ACC (i.e., the immediate precursor of ET) treatments, suggesting that *NAC1* is not linked to stress in *Arabidopsis* [45]. In addition, in the ET-insensitive *ein2-1* mutant, the salt induction of *AtNAC2* was completely abolished, indicating that EIN2 is absolutely required for the signaling pathway that leads to the *AtNAC2* induction under salt stress. However, the salt induction of the *AtNAC2* gene was not affected by EIN3 (i.e., TF downstream EIN2) and was not dependent upon ABI2, ABI3, or ABI4 and the ABA signaling pathway. On the other hand, AtNAC2 promoted lateral root formation, and salt-induced *AtNAC2* expression was dependent upon the ET and auxin signaling pathways [45]. As in Arabidopsis, rice, and cotton, expression of *SlNAP2* is rapidly ABA-induced, demonstrating conservation of the upstream regulatory pathways that control the ABA-mediated induction of *NAP* genes across monocot and eudicot species [42]. In addition, it should be stated that the NAC TFs possessing the C-terminal transmembrane domain (TMD) are located in the ER.

Recently, by upregulating ABA signaling, it was reported in rice that the membrane-bound *ONAC054* is essential for ABA-induced LS [197]. However, the cleavage of TMD allows ONAC054 (i.e., ONAC054α) to relocate to the nucleus. LOF *onac054* mutants exhibited a functional stay-green phenotype. Interestingly, ONA054α has an alternative splice variant (i.e., nuclear ONAC054β). Furthermore, the transcript level of *ONAC054β* increases at a much faster rate than that of *ONAC054α* [198]. In [197], it was demonstrated that the level of two splice variants is low in the absence of exogenous ABA, and the ONA054α,β level rises in the nucleus upon ABA signal perception. Probably, ABA removes TMD from ONAC054α by currently unidentified proteases. The nuclear pool of ONA054α, β promotes LS through direct activation of both OsABI5 and NYC1 (Chl*b* reductase compounds NON-YELLOW COLORING1) transcription [197,199,200]. More recently, it was proved that (i) dehydration stress controls the cleavage of TMD from *ONAC054*, as observed previously in the ABA treatment [197,200]; (ii) *ONAC054* expression is upregulated by several ABRE binding factors (ABF); and (iii) *ONAC054* overexpression improved the grain yield of rice without causing any growth defect [200]. Collectively, all of the above data demonstrate that the activity of ONAC054, which is important for ABA-induced LS in rice, is minutely controlled by multilayered regulatory processes. Interestingly, nuclear SNAC3 protein confers heat and drought tolerance at the rice vegetative stage through modulation of downstream ROS genes [96]. *SNAC3* was shown to be induced by exogenous ABA. Overexpressing *SNAC3* not only enhanced heat and oxidative tolerance via enhancing the cell membrane stability but also improved drought resistance by reducing water loss. Likewise, *SNAC3* likely regulates not only ROS scavenging but also ROS metabolism. Consistently, silencing *SNAC3* by RNAi caused increased sensitivity to drought, high temperature, salinity, and ABA treatments [12,96]. Data from ABA content, *SNAC3* expression, and response in mutants suggest that *SNAC3* may function mainly in an ABA-independent manner [96], in contrast to some previously reported NAC genes. Finally, last year, two SNAC3-OX lines and two SNAC3-RNAi lines were created and subjected to arsenic stress treatments [201]. Interestingly, *SNAC3* overexpression significantly intensified rice tolerance to arsenic stress and boosted grain yield, while the expression elimination generated a contrary response in both parameters. For further clarification, *SNAC3* overexpression induced the enzymatic antioxidant levels of transgenic rice lines which in turn scavenge ROS, causing decreased oxidative stress and enhanced arsenic stress tolerance [201].

On the other hand, MYC2, MYC3, and MYC4 are targets of JAZ2 [202], a gene expressed primarily in the stomata guard cells that directly regulate the expression of *ANAC19*, *ANAC55*, and *ANAC72*, modulating the degree of stomata opening. JAZ2 degradation and the subsequent activation of MYC2, MYC3, and MYC4, in turn, activates ANAC19, ANAC55, and ANAC72. MYC2 and MYC3 bind to the promoter of *ANAC* genes. When a pathogen is perceived, Arabidopsis plants without *JAZ2* are alerted to the stomatal apparatus closure (i.e., stomatal immunity). This research also supports that *JAZ2* is a repressor of *ANAC* gene expression mediated by MYCs [202,203]. Altogether, (i) *NAC072*, *NAC019*, and *NAC055* may act individually in different stress responses. *NAC019* and *NAC055* are involved in JA and/or ET signaling pathways, whereas *NAC072* has been reported to be associated with the ABA-dependent stress response and to be strongly induced by ABA [195,204]; (ii) the *nac019nac055nac072* triple mutant was slightly more sensitive to ABA or dehydration than single or double mutants [205]; (iii) *NAC072, NAC019*, and *NAC055* genes contain ABRE motifs, and so rapidly respond to exogenous ABA [206]; and (iv) the *NAC072* gene is upregulated by overexpression of *ABF3*, and the LOF of *NAC072* turned out in ABA insensitivity. Meanwhile, the LOF of both *NAC072* and *ABF3* further improves ABA insensitivity. In other words, it is interesting to observe that *NAC072* displays a dual function in ABF3-mediated ABA signaling [207].

The interplay between the ET and auxin signaling pathways with NAC TFs is highly intricate and multifaceted. In a recent study, a mechanism has been elucidated wherein the master regulator of mitochondrial disturbance, ANAC017, activates auxin-related genes along with genes associated with the ET pathway. This mechanism plays a crucial role in the control of mitochondrial stress signaling [208]. Significantly, ANAC017 exerts direct control over components in both the auxin and ET pathways. This dual regulatory role indicates the versatility of ANAC017 in coordinating responses to mitochondrial stress and modulating the signaling pathways associated with auxin and ET. The integration of stress signaling with hormonal pathways is a common feature in plant responses to environmental cues, and ANAC017 appears to be a key player in this regulatory network. The binding of ANAC017 to the promoters of *MKK9* and *ACS2* indicates that enhanced ET biosynthesis, as indicated by the activation of ACS2, is associated with an early response to mitochondrial stress. Intriguingly, the manuscript discusses a model illustrating the roles of ET and auxin in governing the mitochondrial retrograde response (MRR) through the direct activation by ANAC017 [208]. In summary, MKK9 and EIN3 collaborate to induce the mitochondrial retrograde response (MRR) as a protective mechanism against mitochondrial disturbance. Concurrently, the ET-induced auxin biosynthesis pathway leads to an increase in IAA levels, which, in turn, suppresses the MRR. This establishes an intricate feedback loop between ET and auxin, allowing for a finely tuned response to mitigate mitochondrial disorganization. Additionally, it has been demonstrated that the nuclear-localized AtNAC017 positively interacts with xyloglucan endo-transglucosylase/hydrolase (*XTH31*), regulating its expression by directly binding to the *XTH31* promoter region. This interaction further emphasizes the multifaceted roles of AtNAC017 in coordinating cellular responses, including interactions with CW-related processes through XTH31 regulation. Likewise, a groundbreaking discovery indicates that, for the first time, ANAC017 functions upstream of the target gene XTH31 to govern aluminum (Al) tolerance and retention in Arabidopsis [48]. In summary, this study demonstrates that (i) *anac017* mutants exhibit reduced Al accumulation in root CW hemicellulose, specifically xyloglucan; (ii) considering the significance of XTH31, XTH15, and XTH17 in hemicellulose modulation, the authors propose the hypothesis that ANAC017 may also regulate the function of these three proteins; (iii) the expression of *ANAC017* and the accumulation of ANAC017 protein significantly decreased under Al stress; and (iv) it has been established that ANAC017 functions as a transcriptional activator of XTH31. In a broader context, the overarching conclusion is that ANAC017 operates upstream of XTH31, thereby regulating Al tolerance in Arabidopsis [48]. This sheds light on the specific involvement of ANAC017 in the cellular responses to Al stress, potentially contributing to the understanding of plant mechanisms in coping with environmental challenges, particularly those related to metal toxicity.

In tomato, a typical climacteric fruit, both SNAC4 and SNAC9 are stimulated by a brief exposure to ET, playing a positive regulatory role in ET synthesis and contributing to the ripening process of tomatoes [134]. The reduced expression of ET signaling genes observed in fruits with silenced SNAC4/SNAC9 confirms the regulatory role of both TFs in ET production. Notably, the silencing of SNAC9 specifically hampers the expression of *LeACS2*, *LeACS4*, and *LeACO1*, which are crucial components of the ET biosynthesis pathway. This feature emphasizes the significance of SNAC9 in modulating specific genes involved in ET synthesis. In summary, *SNAC4/9* has the ability to bind to the promoter regions of ET biosynthesis genes, exerting a positive influence on the ripening process [134].

### 4.2. The Complexity of NAC TFs In Vivo Is Marked by the Existence of Modules

The concept of transcriptional modules suggests that the TFs can form interconnected and functionally related groups. These modules may involve multiple NAC TFs cooperating or sharing regulatory pathways to collectively influence the expression of specific target genes. The formation of transcriptional modules adds complexity to the regulatory network governed by the NAC TF family, highlighting the intricate nature of their interactions and the potential for coordinated control over various biological processes. Understanding these modules can provide valuable insights into the precise regulatory mechanisms orchestrated by NAC TFs in different cellular contexts. However, the study of in vivo complexes remains to be addressed. Chromatin immunoprecipitation (ChIP) followed by high-throughput sequencing (ChIP-Seq) is a powerful technique that significantly advances our understanding of these transcriptional modules. It allows researchers to identify and map the genomic locations where specific TFs bind, providing critical insights into the regulatory elements that control gene expression and contribute to the complexity of transcriptional networks. This emphasizes the crucial role of the *AtNAC017*, *AtNAC082*, and *AtNAC090* modules in modulating the timing of LS in Arabidopsis [83]. The regulatory landscape of LS involves multiple players, and *AtNAC042* has been identified as a key negative regulator that modulates cellular H_2_O_2_ levels. It is noteworthy that while AtNAC017 mediates responses to H_2_O_2_ in plants, it is not induced by H_2_O_2_ itself. Intriguingly, the loss of ANAC017 function leads to increased ROS levels, contributing to the acceleration of LS. This underscores the complexity of regulatory interactions within the plant senescence process, where different NAC TFs play distinct roles in modulating cellular responses to oxidative stress [83]. Despite *AtNAC090* not being induced by SA treatment, both *AtNAC090* and *AtNAC017* assume predominant roles in regulating SA and ROS responses within the *AtNAC017*, *AtNAC082*, and *AtNAC090* module [83]. The distinct roles of AtNAC090 and AtNAC017 within the AtNAC017, AtNAC082, and AtNAC090 module, particularly in regulating SA and ROS responses, further highlight the functional diversity and specificity of individual NAC TFs. Unfortunately, despite a considerable amount of data on the *AtNAC017*-*AtNAC082*-*AtNAC090* module, it is unknown how it performs its role once bound to DNA in vivo. Understanding modules at this level is a significant challenge for the future.

The role of *TgNAP* in *Tulipa gesneriana* contrasts with its counterpart *AtNAP* in Arabidopsis. While *AtNAP* induces precocious LS, TgNAP promotes petal senescence in *Tulipa gesneriana*. *TgNAP* achieves this by activating the expression of genes involved in SA biosynthesis, such as *TgPAL1* and *TgICS1*. Additionally, *TgNAP* enhances pathways related to the detoxification of ROS. This illustrates the versatility of NAC TFs in different plant species and highlights their capacity to regulate diverse biological processes, including senescence, in a context-dependent manner. This illustrates the diverse functions of NAP-like TFs in regulating senescence processes in different plant tissues and species [209]. It effectively highlights the dual and multifaceted roles of NAC TFs in the regulation of LS, acting as both positive and negative regulators and influencing the process through the modulation of SA and ROS pathways. The intricacies of these regulatory networks demonstrate the versatility of NAC TFs in orchestrating the complex processes associated with plant senescence. In reference [209], it was revealed that TgNAP serves as a positive regulator in H_2_O_2_-induced LS, and SA induces the expression of *TgNAP*. This implies that *TgNAP* plays a role in both H_2_O_2_-induced LS and SA biosynthesis in tulip, emphasizing its involvement in the complex regulatory network governing senescence processes.

Returning to the functional module formed by *AtNAC017*, *AtNAC082*, and *AtNAC090*, it is noteworthy that these three components interact with and directly regulate the promoters of target genes [83]. This study reveals variations in regulatory importance among the components of the module. Specifically, *AtNAC090* emerges as the primary regulator in downregulating LS by SA and ROS responses. In contrast, *AtNAC017* takes on a prominent role in ROS eradication. This differential contribution emphasizes the nuanced and specialized functions of each component within the regulatory module.

As previously mentioned, JAZ2 primarily functions within guard cells. In contrast, the roles of *ANAC19*, *ANAC55*, and *ANAC72* are not limited to guard cells; they extend to mesophyll cells where these proteins act to repress SA defenses. This highlights the diverse cellular functions and regulatory activities exhibited by different components within the plant defense and signaling pathways [210]. The findings highlight the presence of a signaling module, consisting of *COI1-JAZ2-MYC2,3,4-ANAC19,55,72*, responsible for the regulation of stomatal aperture. Notably, CORONATINE-INSENSITIVE-1 (COI1) interacts with JAZ proteins, facilitating JAZ ubiquitination and degradation through the 26S proteasome pathway. In summary, bacterial infection facilitated by CORONATINE induces the degradation of JAZ2, leading to the subsequent activation of *MYC2*, *MYC3*, and *MYC4*. These activated MYC proteins, in turn, transcriptionally induce the expression of *ANAC19*, *ANAC55*, and *ANAC72*, contributing to the overall regulatory response in plants. In *Nelumbo nucifera* Geartn, specific NAC TFs, namely *NnNAC45*, *NnNAC003*, *NnNAC016*, *NnNAC043*, *NnNAC060*, and *NnNAC070*, demonstrate significant induction under NaCl treatment. Notably, their functions in response to salt stress are contingent upon ABA signaling, suggesting that ABA plays a regulatory role in the functions of these NAC TFs in the context of salt stress. The induction of these NAC genes underlines their potential involvement in the plant’s adaptive mechanisms to cope with saline conditions [59]. The ability of NAC016, NAC025, and NAC070 proteins to form complexes with other proteins potentially indicates a collaborative role in plant processes [59]. Knockout of *OsNAC041* alters plant hormone homeostasis, possibly causing the salt-sensitive phenotype [130]. In apple, the *MdNAC047* gene was isolated and functionally characterized for its involvement in ET regulation of salt tolerance. *MdNAC047* overexpression facilitated the release of ET and increased the expression of ET-responsive genes [211].

Studying *SNAC5*, *SNAC6*, *SNAC9*, and *SNAC10* genes and utilizing RNA-Seq and ChIP-Seq, Chung et al. (2018) identified 391 direct target genes. Among these, 239 loci were bound by *SNAC5*, *SNAC6*, *SNAC9*, and *SNAC01,* leading to the upregulation of 65, 91, 115, and 186 genes, respectively [212]. On the other hand, *AaNAC1* from *Artemisia annua* (sweet wormwood; Chinese wormwood) was induced by SA and MJ, indicating its potential roles in abiotic or biotic stresses in *A. annua* [213]. Specifically, SA induces *AaNAC1* expression about 30-fold, while MJ induces a 40-fold increase. Under the treatment of SA + MJ, the expression level of *AaNAC1* can reach up to 60-fold. The findings regarding *AaNAC1* suggest its significance in improving artemisinin content, which is an antimalarial drug localized in trichomes, as well as enhancing plant defense. The study’s conclusions propose that *AaNAC1* has potential applications in transgenic breeding to enhance both artemisinin content and drought tolerance in *Artemisia annua*. This highlights the potential of NAC TFs, such as *AaNAC1*, in molecular breeding strategies for the development of crops with improved pharmaceutical and stress tolerance traits [214]. Interestingly, a current study provides the first indication of the molecular basis of artemisinin biosynthesis regulation through YABBY–WRKY interactions, regulated by AaJAZ8. Additionally, it highlights the potential of *AaYABBY5* overexpression plants as a valuable genetic resource for artemisinin biosynthesis [215].

Several studies showed that plant-specific *NAC103* TF has a variety of biological functions in various stress responses. Recently, *NAC103* was reported to be involved in the expression of genes related to DNA damage response [111]. Genetic experiments demonstrated that *NAC103* is a positive regulator in the ABA pathway and also regulates several ABA-responsive downstream genes. *NAC103* is upregulated by ABA treatment at both the transcriptional and post-translational levels and plays an important role in ABA responses, especially during seed germination. Overexpression of *NAC103* inhibited seedling growth when plants were treated with ABA.

## 5. Spatial Expression of *NAC* Genes

Studying the spatiotemporal expression patterns provides insights into when and where specific NAC genes are active, shedding light on their contributions to plant growth, adaptation to environmental challenges, and other important biological processes i.e., [4,69,216,217]. Let us take a look at some outstanding works. The observation that the ANAC017-mRNA level is higher in flowers and siliques (i.e., seed formation) and relatively lower expression in roots, leaves, and stems suggests that this NAC TF may play specific roles in reproductive organs [48]. The study by Li et al. (2021) [181] that analyzed 80 *HaNAC* genes in *Helianthus annuus* (sunflower) provides valuable information about the expression patterns of these genes in various plant organs. Notably, the expression analysis across different organs, including pistil, stamen, mature leaves, roots, and seeds, sheds light on the tissue-specific roles of *HaNAC* genes. The observation that, except for *HaNAC23* and *HaNAC105*, the remaining 78 genes were lowly expressed in seeds suggests that these particular *NAC* families may not play a prominent role during seed development or maturation [181]. The tissue-specific expression pattern of *ONAC127* and *ONAC129* in rice, with predominant expression in the pericarp and weaker expression in the starch endosperm and aleurone layer, proposes a role in seed development, particularly during the early and middle stages [52]. Understanding the specific functions of *ONAC127* and *ONAC129* in the pericarp and their potential interaction as a heterodimer could provide insights into the regulatory mechanisms of seed development. The findings regarding *ClNAC68* provide valuable insights into its role in seed development and germination in watermelon (*Citrullus lanatus*) [177]. The knockout of *ClNAC68* is associated with delayed seed maturation and inhibited germination. The observed decrease in IAA and ABA contents in seeds of *clnac68* mutants during development suggests the involvement of *ClNAC68* in regulating phytohormone levels. Specifically, this study indicates that *ClNAC68* was highly expressed in sweet flesh and acts as a transcriptional repressor that positively regulates seed germination. This positive regulation is achieved by increasing the free-IAA content through the direct repression of the IAA deactivator *ClGH3.6*. Additionally, *ClNAC68* is implicated in the positive regulation of *ABI5* expression, enhancing ABA sensitivity and influencing seed germination. These findings highlight the complex regulatory mechanisms involving *ClNAC68* in the control of phytohormone levels and the expression of key genes related to seed germination and maturation in watermelon [177,178]. The high expression of *OsNAC24* in the immature endosperm of rice grains may indicate its involvement in the control of processes such as nutrient allocation, storage compound biosynthesis, and overall seed development. The specific expression pattern of *OsNAC24* in the immature endosperm points to its likely involvement in the molecular pathways associated with endosperm maturation, nutrient storage, and possibly other aspects of seed development [218]. Interestingly, ChIP-PCR assays revealed that the *OsNAC24* gene directly targets the promoters of six starch-synthesis enzyme-coding genes and regulates their expression. The *OsNAP* gene interacts with *OsNAC24* to coactivate the expression of its target genes and facilitates the localization of *OsNAC24* to the nucleus. In other words, OsNAP and OsNAC24 likely form a protein–protein interaction that enhances their ability to regulate specific target genes [218].

A detailed study on the spatial expression of NAC genes was recently published in an article about jojoba (*Simmondsia chinensis*) [219]. It is interesting to note that a total of 57 NAC genes have been identified. Among these, seven are highly expressed in seeds, and specifically, *ScNAC34*, *ScNAC37*, and *ScNAC14* show higher expression in the embryonic axis compared to other genes, suggesting potential key roles in seed development. Additionally, *ScNAC34*, *ScNAC37*, and *ScNAC39* are highly expressed in both cotyledon and embryonic axis tissues. Indeed, the positive or negative correlation between the expression of *NAC* genes and seed development suggests that *ScNAC* genes likely play a regulatory role in the growth and development of *Simmondsia chinensis* seeds. Moreover, as previously determined for the *ClNAC68* gene (210), the presence of auxin response elements in 30 *ScNAC* genes indicates their potential involvement in auxin-mediated processes during the development of jojoba seeds [219].

In relation to *Triticum aestivum*, quantitative PCR (qPCR) analysis revealed higher expression levels of *TaNAC100* in developing seeds, particularly at 5 DAF (days after flowering). This suggests that *TaNAC100* may play a specific role during the early stages of *Triticum aestivum* seed development. The increased expression of *TaNAC100* in developing seeds, as opposed to other tissues such as stems, leaves, roots, and young spikes, indicates its potential involvement in processes related to seed formation and maturation [185]. The observation that *TaNAC100* demonstrates an appropriate spatiotemporal expression pattern and co-localizes with GLU-1 (a gene associated with glutenin synthesis in wheat) suggests the possibility of functional interactions or shared regulatory pathways between these two genes. This feature could potentially contribute to the overall quality and characteristics of wheat seeds. To gain deeper insights, it would be beneficial to investigate the specific functions of *TaNAC100* and *GLU-1*, explore their regulatory networks, and determine whether they directly interact or influence each other’s expression. The findings from Li et al. (2021) indicate that *TaNAC100* functions as a transcriptional repressor in modulating *GLU-1* expression [185]. Notably, similar to *OsNAC2* [123], overexpression of *TaNAC100* also leads to a significant reduction in plant height and a delay in flowering time, suggesting a conserved function between *TaNAC100* and *OsNAC2*. Further investigations into the specific mechanisms through which *TaNAC100* and *OsNAC2* regulate plant height and flowering time would be intriguing. Additionally, exploring their common downstream targets or pathways could provide valuable insights into their shared regulatory functions.

In soybean, a recent study has discovered two NAC TFs exclusively expressed in nodules and endosperm, along with one that is spatiotemporally expressed. Firstly, SNAP1-4 NAC TFs, referred to as Soybean Nitrogen Associated NAPs (SNAPs), primarily impact the N-responsive transcriptome in mature nodules and rank among the most highly connected hub TFs. Secondly, NAP clade TFs exhibit evolutionary conservation. The authors of this study identified 388 genes directly regulated by SNAP1-4 that responded to nitrogen (N) (223 activated and 165 repressed) [180]. The nodules of *snap1-4* quadruple mutants exhibit reduced sensitivity to the high N inhibition of nitrogenase activity, leading to an acceleration of senescence. Additionally, RT-qPCR analysis confirmed a significant reduction in mRNA levels for all target genes in the nodules of these mutants. Integrative analysis reveals that *SNAP1*-*4* significantly impacts the transcriptional response to high N through the direct regulation of a subnetwork comprising senescence-associated genes and transcriptional regulators. It is proposed that the transcriptional network mediated by *SNAP1*-*4* may trigger nodule senescence in response to high N [180]. Intriguingly, *SNAP1*-*4* genes play pivotal roles in regulating nodule responses to high N, and the mature nodules of *snap1*-*4* quadruple mutants demonstrate a notable tolerance to high N exposure.

Secondly, an important function of the NAC family has recently been proposed in endospermic seeds [220]. Previously, it was demonstrated that the endosperm-specific ZOU (bHLH-type TF) promotes the weakening of endosperm CWs and facilitates invasive embryo growth [188]. The weakening of CWs likely occurs through the activation of CW-modifying enzymes [221]. Consequently, in *ZOU* mutants, the elimination of endosperm is delayed, leading to significant damage to embryo growth. Intriguingly, the role of *ZOU* in activating PCD seems to be evolutionarily conserved [222]. The gradual elimination of the endosperm in WT is initiated by a PCD program, likely regulated by multiple NACs. The specific NACs involved in this process remain unidentified [223]. The work by Doll demonstrates that both ZOU-controlled CW weakening and the PCD promoted by both *ZOU* and NACs are essential. These processes work together to eliminate the mechanical resistance of the endosperm, facilitating the rapid and effective invasive growth of the *A. thaliana* embryo. In essence, (i) *ZOU* is necessary for initiating PCD, and (ii) there are two parallel pathways: ZOU regulates endosperm softening, while PCD regulates endosperm loosening. As a result, the combined action of CW weakening and PCD actively facilitates the breakthrough of the embryo through the active auto-elimination of endosperm cells [220].

Thirdly, the NAC103 protein is a substrate of the 26S proteasome and is maintained at a low concentration under normal conditions [110]. Nuclear *NAC103* can form hetero-complexes with various other NAC TFs. *NAC103* is expressed in two sites (i.e., seeds and seedlings) and plays an important role in ABA responses, especially during Arabidopsis seed germination. Its expression is significantly upregulated by ABA, with the encoded protein being stabilized by exogenous ABA [110,224]. Under ABA treatment, overexpression of *NAC103* inhibits seed germination rates and seedling growth, while *nac103* mutants display increased seed germination rates. Previous reports indicate interactions between ABA-responsive NAC proteins and RING-H2, suggesting that this interaction regulates the expression of downstream genes [224]. All of these results suggest that the structure of NAC103 in the N-terminus may play a role in regulating ABA ABA downstream genes. These results suggest the possibility that the overaccumulation of NAC103-mRNA is responsible for root growth inhibition under conditions of excess boron in the *rpt5a-6* mutant. Certainly, the mutation in *NAC103* (*rpt5a-6*) alleviates DNA damage in an *A. thaliana* mutant sensitive to excess boron [179].

## 6. Involvement of *NAC*s in Seed Development, Dormancy, and Germination

As previously indicated, NAC TFs play notable roles during plant growth and development, exerting significant influence on stress processes. However, the specific role of NACs in the first phases of the seed cycle has not yet been adequately addressed. Nevertheless, *NARS1* and *NARS2* regulate embryogenesis in *A. thaliana* by controlling ovule integument development and degeneration, seed morphogenesis, and silique senescence [225]. These authors provide evidence that (i) both TFs regulated the development of cells in the outer integument; (ii) the development of endosperm is delayed in *nars1 nars2* mutant; (iii) *NARS1* and *NARS2* are not expressed in the torpedo-shaped embryo stage; (iv) the absence of both genes in the embryo does not affect embryogenesis; and (v) *NARS1* and *NARS2* redundantly regulate not only seed morphogenesis but also silique-specific processes [225].

### 6.1. Involvement of NAC TFs in Seed Reserve Accumulation

The function(s) of NAC TFs in the endosperm, constituting approximately 90% of kernel weight, have not been fully elucidated. Therefore, the molecular mechanism underlying nutrient uptake and the biosynthesis of storage reserves involved in endospermic seed filling, particularly in well-studied species of grasses such as rice and maize, has been recently updated [226]. The size and weight of seeds are critical characteristics in agronomically exploitable species, playing pivotal roles in determining their agricultural value and overall utility. In rice, the conserved genes *OsNAC020*, *OsNAC023*, and *OsNAC026* are primarily highly expressed during seed development, and are specifically associated with grain (caryopsis) size and weight [227]. *OsNAC020* and *OsNAC023* have the capability to form dimers with *OsNAC026*. Additionally, these dimers are reported to localize in the nucleus. Possibly, through their dimerization with *OsNAC026*, *OsNAC020* and *OsNAC023* are involved in the regulation of genes associated with seed size and weight [132,227]. On the other hand, *ONAC127* and *ONAC129* are expressed in the caryopsis and are involved in the regulation of starch accumulation and carbohydrate transport [52].

Nearly 30 genes participate in starch synthesis in maize [97]. However, the mechanism by which starch biosynthesis is regulated remains unclear, and specific information about the direct involvement of NAC TFs in their regulation is limited [44]. Both ZmNAC36 and ZmNAC34 are implicated in directly or indirectly regulating starch synthesis [58,189,190]. The *ZmNAC34* gene, a classic TF with nuclear localization and transactivation activity, is specifically expressed during maize endosperm development, exclusively at 15 and 20 DAP. Its pivotal role in endosperm function is well-established [58]. *ZmNAC34’*s impact extends beyond maize, as evidenced by studies in rice. Overexpression of *ZmNAC34* in rice resulted in decreased soluble sugar content and total starch accumulation. This effect was attributed to the downregulation of starch biosynthesis-related genes, leading to alterations in starch spatial structure [58]. In other words, *ZmNAC34* is regarded as a negative regulator of starch biosynthesis, inhibiting the process. On the other hand, *ZmaNAC36*, encoding a predicted 38.40 kDa protein, exhibits strong expression during maize endosperm development, peaking at 15 DAP, compared to roots, stems, embryos, and flowers. *ZmaNAC36* is co-expressed with most starch synthetic genes. Transient overexpression of *ZmaNAC36* led to the upregulation of numerous starch synthesis genes in the endosperm, highlighting its regulatory role [189]. In conclusion, the close phylogenetic relationship between *ZmNAC34* and *ZmNAC36*, both involved in the regulation of starch biosynthesis genes [189], raises the question of whether the function of *ZmNAC34* in rice mirrors its function in maize, a topic that remains to be elucidated. In addition to *ZmNAC36*, the redundant genes *ZmNAC128* and *ZmNAC130*, whose proteins do not interact, have been identified as positive regulatory factors for starch biosynthesis-related genes in rice and maize [44,228].

On the other hand, two functionally redundant TFs, named *ZmNAC128* and *ZmNAC130*, not only act as activators of zeins in *Zea mays* but are also specifically and strongly expressed during the filling stage [44]. Deficient expression of *ZmNAC128* and *ZmNAC130* leads to a decrease in starch and protein accumulation. These genes directly regulate the expression of *16-kD γ-zein* and *AGPS2* (Bt2) by binding to a specific cis-element (ACGCAA) in their promoters [44]. Therefore, *ZmNAC128* and *ZmNAC130* are essential for maize endosperm filling, and both genes cooperate with the bZIP TF OPAQUE-2 (O2) promoter [191]. However, the specific contribution of each of the *ZmNAC128* and *ZmNAC130* genes to endosperm filling remains largely unknown. Recent results from Chen et al. (2023) [191] clearly indicate that *ZmNAC128* and *ZmNAC130* play crucial roles in the biosynthesis of zeins and starch, redundantly promoting endosperm filling. Their simultaneous LOF severely impaired the accumulation of storage reserves. In addition to *16-KD*, the genes *27-kD* and *50-kD γ-zein* are also direct targets of *ZmNAC128* and *ZmNAC130*. Interestingly, *ZmNAC128* and *ZmNAC130* also recognize cis-elements beyond the ACGCAA motif [191]. Knocking down the expression of *ZmNAC128* and *ZmNAC130* through RNA interference (RNAi) resulted in a shrunken kernel phenotype with a significant reduction in starch and protein content [44]. The absence of *ZmNAC128* and *ZmNAC130* had a pleiotropic effect on the utilization of carbohydrates and amino acids [44,191]. *ZmNAC130* interacts with *ZmNAC128*, but they do not dimerize with each other. Both genes exhibit functional redundancy as they regulate the expression of the same downstream genes, such as 16-KD γ-zein and Bt2 [44,191]. In conclusion, the endosperm-specific *ZmNAC128* and *ZmNAC130* genes coordinate the accumulation of starch and proteins by regulating the expression of key starch biosynthetic enzymes and major seed proteins, including AGPS2 (Bt2) and the 16-kDa γ-zein (a type of prolamin). In other words, both genes oversee the rate-limiting step of starch synthesis [44].

In addition, the gene *NAC-A18*, predominately expressed in developing grains, regulates both starch and storage protein synthesis. This gene has been identified to directly bind to the cis-element ACGCAA in the promoters of *Triticum aestivum TaLMW-D6* and *TaLMW-D1* [41]. Ectopic expression of wheat *NAC-A18* in rice significantly decreased starch accumulation and increased seed storage proteins (SSP) accumulation. In wheat, the nucleus-localized transcriptional repressor *TaNAC019-A1*, homologous to *HvNAC019* [153], plays a crucial role in endosperm starch synthesis and negatively regulates kernel size and weight. Overexpression of *TaNAC019-A1* inhibits starch synthesis in both wheat and rice kernels, directly binding to the promoters of *TaAGPS1-A1* and *TaAGPS1-B1* genes, thereby suppressing their expression. This repression of starch synthesis is achieved by downregulating the expression of multiple genes involved in starch synthesis, including *TaAGPS1a* and *TaAGPS1b* [43]. However, there is conflicting evidence, as Gao et al. (2021) [186] demonstrated that *TaNAC019* acts as a promoter of starch synthesis and enhances the expression of SSP genes. Similarly, Li et al. (2021) [46] found that the *TaNAC100* gene has the highest expression among all organs in developing seeds, peaking at 10 DAP, while *TaNAC020* peaks at 15 DAP. *TaNAC020*s are predominantly expressed in developing grains and play a positive role in starch synthesis and accumulation. They are key regulators of seed size and number. Finishing the update of the notable *T. aestivum* NAC TFs, *TaNAC100* functions as a transcriptional repressor, modulating *GLU-1* expression by binding to its conserved cis-regulatory module (CCRM). Additionally, *TaNAC100* is involved in the crosstalk between protein and starch synthesis in *T. aestivum* seeds [46].

The synthesis of SSPs and starch is spatiotemporally regulated at the transcriptional level by synergistic interactions between TFs and *cis*-elements distributed in SSP gene promoters. Recently, the *TaSPR-A*, *TaSPR-B*, and *TaSPR-D NAC* genes were found to be preferentially expressed in developing endosperms and were reported to suppress the transcription of SSP genes in *T. aestivum*, which has a complex genome [187]. The novel *TuSPR* gene was also found to be preferentially expressed in the developing endosperm during *T. urartu* filling, a species with a simpler genome. Both *TuSPR* and *TaSPR*, initially characterized as transcriptional repressors of SSP synthesis in *T. aestivum*, directly regulate SSP genes. Interestingly, in *TuSPR*-overexpressing common wheat, both the transcription of SSP genes and the accumulation of SSPs were impeded. *TuSPR* regulates *SSP* genes directly and binds to SSP gene promoters. *TuSPR* interacted with no *TaSPR-A*, *TaSPR-B*, or *TaSPR-D* genes [187].

Four NAC proteins—ZmNAC128 and ZmNAC130 from maize, and OsNAC20 and OsNAC26 from rice—sharing a high sequence similarity, were found to bind to a common sequence, ACGCAA, coordinating starch and protein synthesis [44,229]. OsNAC20 and OsNAC26 directly influence starch and storage protein accumulation by regulating the expression of genes related to their synthesis. In a study conducted by Wang et al. [229], it was demonstrated that OsNAC20 and OsNAC26 directly bound to the promoters of *GluA1*, *GluB4*, a-globulin, and 16 kD prolamine, regulating storage protein synthesis. Additionally, OsNAC20 physically interacts with OsNAC26 in plant cell nuclei. The NAC domains of OsNAC20 and OsNAC26 bind to the same *cis*-elements, activating similar starch and protein synthase-related genes. Similar to maize, the reduced starch content in the *osnac20/26-1* mutant is not associated with the low expression of starch synthesis-related genes *AGPS2b* and *AGPL2*. In summary, OsNAC20 and OsNAC26 play an essential and redundant role in regulating starch and storage protein synthesis in rice [229]. Loss of function (LOF) of OsNAP led to altered expression in all tested starch and protein synthesis-related genes and reduced starch content. As mentioned earlier, CACG constitutes the core NAC-binding motif [30,204]. Jin’s research revealed that in addition to CACG, other motifs like TTGACAA, AGAAGA, and ACAAGA are present in the promoter regions of each of the *OsNAC24* target genes [218].

In watermelon, the biological function of *ClNAC68* has been investigated, revealing that this transcriptional repressor plays a critical role in sugar accumulation and seed development [178]. *ClNAC68* positively regulates the accumulation of sugars, particularly sucrose, fructose, and glucose, by influencing the contents of free-IAA and -ABA. These findings align with the lower sucrose content observed in *clnac68* mutant lines, where three SWEETs (sugar transporter family) were significantly downregulated. Additionally, *ClNAC68* regulates sugar content by repressing invertase activity in watermelon [179]. In an interesting parallel, two recent studies, unrelated to the NAC family, demonstrated that *ZmABI19* (ortholog of FUS3) and *ZmbZIP29* are highly expressed in the early stage of seed development, playing initial roles in seed filling, and their expression is induced by ABA [87,175]. Overexpression of *ZmABI19* and *ZmbZIP29* resulted in enhanced *O2* expression and increased seed weight and the contents of starch and zein proteins. As mentioned earlier, the endosperm-specific *O2* is considered a central regulator of protein and starch synthesis. As ABA contents substantially increase at 8 DAP, endosperm filling is initiated. Thus, the elevated ABA levels promote *O2* expression during endosperm filling. In other words, *ZmbZIP29* is an ABA-inducible TF and directly regulates *O2* expression genetically in the endosperm. The level of ZmbZIP29-mRNA is generally lower in the endosperm than in the embryo. ZmbZIP29 interacts with ZmABI19, exhibiting an additive transactivation pattern. Notably, the phosphorylation of ZmABI19, ZmbZIP29, and O2 by SnRK2.2 is essential for their full transactivation capacity. Cumulatively, mutations in *zmbzip29* and *zmabi19* significantly influence seed development and storage protein synthesis [87,175]. Taken together, *ZmNAC128*, *ZmNAC130*, *ZmABI19*, and *ZmbZIP29* cooperate with *O2* to facilitate endosperm filling and the synthesis of zeins and starch in the starchy endosperm [87,175,191]. That is, the ABA-mediated phosphorylation of two central coordinators, ZmABI19 and ZmbZIP29, by SnRK2.2 enhances gene transactivation for endosperm filling.

### 6.2. The Participation of NAC TFs in Seed Dormancy and Germination Processes

Seed dormancy is an omnipresent phenomenon in plants, induced and controlled by ABA [230,231,232]. Vivipary, characterized by low levels of dormancy, can lead to preharvest sprouting and substantial economic losses [233]. Upon the disappearance of dormancy under suitable conditions, germination ensues. Seed germination marks the initiation of the second cycle in a plant’s life, constituting a plant-specific developmental process wherein an embryonic plant grows to form a seedling. While the majority of prior studies have focused on NAC-mediated regulation of stress and LS, only a few have delved into the involvement of NACs in seed dormancy and germination.

As discussed earlier, *OsNAC2* plays a crucial role in rice development and abiotic responses by binding to various downstream targets [125]. It inhibits seed germination and coleoptile growth, with *OsACO* or *OsABA8ox1* operating downstream of *OsNAC2*. In essence, *OsNAC2* acts as a regulatory element in ET, GAs, and ABA signaling, serving as a metabolic component to modulate rice seedling growth. The inhibition of seed germination by *OsNAC2* is likely through the ABA pathway, targeting the promoters of *OsNCED3*, *OsABA8ox1*, and *OsZEP1* [234]. The promotive effect of ET on seed germination may be counteracted by the excessive accumulation of ABA. Recent findings underscore the role of *OsNAC2* in regulating seed dormancy release and the germination process [11]. *OsNAC2*, acting downstream of ABA signaling and upstream of GA signaling, exhibits high expression during seed development and germination compared to other plant tissues. *OsNAC2* overexpression induces (i) inhibition of ABA degradation, leading to increased ABA content during early germination; (ii) enhanced seed dormancy and suppression of germination; (iii) the necessity of *OsNAC2* for maintaining primary seed dormancy; and (iv) direct binding of *OsNAC2* to the promoters of ABA metabolism genes (i.e., *OsABA8ox1*, *OsABA8ox2*, and *OsABA8ox3*), inhibiting their transcription during seed germination [11]. Notably, *OsNAC2* is expressed in dry seeds, with its mRNA markedly diminishing during germination. In summary, experiments in [11] provide evidence supporting the hypothesis that *OsNAC2* is involved in the regulation of ABA metabolism to maintain primary seed dormancy.

Recently [227], it was conjectured that *OsNAC20* and *OsNAC26*, whose mRNAs are transcribed in the very early stages of rice zygotic embryogenesis, might function in controlling seed dormancy, as both genes are exclusively detected in the developing endosperm. Alongside *OsNAC20* and *OsNAC26*, eight other NAC genes were identified in *O. sativa* seeds. Notably, *OsNAC25*, *OsNAC41*, *OsNAC128*, and *OsNAC129* were found to be downregulated in the endosperm of the *osnac20/26* mutant. Among these, *OsNAC25*, *OsNAC128*, and *OsNAC129* are endosperm-specific genes, with *OsNAC25* exhibiting a similar expression pattern to that of *OsNAC20* and *OsNAC26* [227]. Interestingly, *OsNAC127* and *OsNAC129* were identified as regulators of the seed-filling process, forming heterodimers. Both overexpression and knockdown of these genes resulted in incomplete grain filling and shrunken grains. OsEATB (AP2/ERF factor), OsMSR2 (calmodulin-like protein), OsSWEET4 (sugar transporter), and OsMST6 (monosaccharide transporter) were identified as direct targets of *OsNAC127* and *OsNAC129* proteins during rice grain filling [52]. In summary, data from Ren et al. (2021) suggest that *OsNAC127* and *OsNAC129* might play critical roles in the translocation and mobilization of starch to the developing rice endosperm [52].

Rice *OsNAC041*, induced by salinity, plays a role in affecting seed germination under salt stress [130]. Additionally, (i) several hormone-associated pathways contribute to *OsNAC041*-mediated salt tolerance; (ii) *OsNAC041* influences ABC transporters, thereby regulating growth and development; and (iii) *OsNAC041* is associated with the ROS system and membrane protection, modulating salt sensitivity [130]. On a different note, *OsNAC3* positively regulates seed germination by involving the ABA pathway and the cell expansion gene *OsEXP4* [12]. Consequently, *OsABA8ox1* and *OsEXP4* are considered putative target genes of *OsNAC3*. In other words, *OsNAC3* directly binds to the promoters of *OsABA8ox1* and *OsEXP4*, inducing their expression during seed germination. This implies that *OsNAC3* is actively involved in cell expansion during seed germination through ABA signaling. Notably, *OsNAC3* expression is not detected in dry rice seeds; it, along with OsEXP4, peaks sharply at 12 h of imbibition, declining thereafter [12]. Acting as a nuclear protein, OsNAC3 directly influences embryo cell elongation by regulating *OsEXP4* expression. In conclusion, the expressions of *OsABA8ox1* and *OsEXP4* are activated during rice seed germination [12].

A T-DNA insertion in the *ANAC060* gene results in increased seed dormancy compared with the WT [235]. Recently, Song et al. (2022) conducted a comprehensive study of *ANAC040*, *ANAC060*, and *ANAC089* [236]. *ANAC040* is likely more ancestral than *ANAC060* and *ANAC089*. These genes, located in the same synteny cluster, indicate a common genomic origin shared with other eudicots but lacking in monocots. This was not expressed in dry seeds. Mutant *anac060-1* and *anac060-2* seeds exhibited deeper primary dormancy. The dormancy of *anac060-2 anac040* did not significantly differ from *anac060*, suggesting that *ANAC060* is epistatic over *ANAC040* in dormancy regulation. *ANAC040* is epistatic over *ANAC060* in regulating salt sensitivity, as *anac040* and *anac060-1 anac040* were resistant to salinity, whereas Col-0 and *anac060-1* were not. However, functional redundancy exists between *ANAC060* and *ANAC040*, but not between *ANAC060* and *ANAC089* [236]. *ANAC060* regulates seed dormancy and sugar (glucose and fructose) sensitivity [234]. Notably, *ANAC060* expression levels are high in dry seeds but significantly reduced during seed imbibition. Additionally, *ANAC060* expression is highly induced by cold stratification. *ANAC060* and *ANAC040* have opposite functions compared to *DOG1*, with DOG1 being crucial for inducing seed dormancy [237], whereas *ANAC040* and *ANAC060* inhibit seed dormancy.

Regarding *ANAC089*, its expression occurs during seed maturation, peaking in dry seeds and decreasing during imbibition [238]. The *gap1-2* mutant reduces the transcript abundance of *ANAC089*, and the *GA3ox2* gene responsible for active GA biosynthesis is induced in the *gap1-2* mutant, suggesting increased levels of bioactive GAs to promote germination. Alterations in the cell redox status translocate *ANAC089* into the nucleus and promote ANAC089 protein accumulation. ABA can induce the expression of *ANAC089* and protein accumulation in seeds and seedlings. The authors conclude that *ANAC089* regulates NO levels and cell redox, represses ABA synthesis and signaling, and binds specifically to genes controlling seed germination and abiotic stress [238]. Remarkably, redox-sensitive *ANAC089* integrates ABA and NO responses during seed germination, conferring key functions to this central regulator in the successful growth of the plant under abiotic stress.

In a groundbreaking study, Kim et al. (2008) demonstrated that NAC with TRANSMEMBRANE motif 1-LIKE 8 (*NTL8* or *ANAC040*), an Arabidopsis membrane-bound NAC TF, accumulates in response to growth and temperature, specifically induced by high salinity [239]. Recent advancements in our understanding of plant cold response, as outlined in [57], have shed light on the evolving roles of NAC TFs. *NTL8* is significantly induced during imbibition and further elevated by cold treatment, suggesting that *NTL8* expression is closely associated with the early stages of seed germination. Recent studies have prominently highlighted the central role of *NTL8* as a cold-specific transcriptional activator in orchestrating plant cold response. This key regulatory function underscores *NTL8’*s pivotal role in governing molecular processes that contribute to plants’ ability to adapt and thrive under cold conditions [57]. NTL8 undergoes controlled proteolytic release from its membrane-bound form, and this release is specifically activated by high salinity. This release allows *NTL8* to function as a transcriptional activator in the nucleus, where it can regulate the expression of target genes involved in various processes, including seed germination. Moreover, in the cold-resistant mangrove species *Kandelia obovata*, a recent study investigated the expression patterns of *KoNAC6*, *KoNAC15*, *KoNAC20*, *KoNAC38*, and *KoNAC51* genes under cold treatment. The findings strongly underscore the vital role played by *KoNAC* genes in the adaptation and response of *Kandelia obovata* to chilling conditions [168]. This insight enhances our understanding of the molecular mechanisms underlying cold resistance in plants. *NTL8* acts by sharply repressing *GA3ox1*, thereby implicating itself in the salt regulation of seed germination through the GAs pathway. Essentially, *NTL8* reduces seed germination under salinity by modulating the GA signaling pathway, which typically promotes germination under favorable growth conditions [239]. In other words, GAs repress both *NTL8* transcription and NTL8 processing. Moreover, *NTL8* may function as a molecular link that incorporates environmental signals into the GA-mediated signaling pathway in seed germination. The *ntl8-1* mutant, on the other hand, functions as a positive regulator of seed germination under high salt conditions. Importantly, *NTL8*-mediated salt signaling is found to be independent of ABA [239]. Further insights into *NTL8’*s role come from Tian et al. (2017), who identified TRIPTYCHON (*TRY*) and TRICHOMELESS1 (*TCL1*) as direct targets of *NTL8* in Arabidopsis. *NTL8* negatively regulates trichome formation by directly activating the expression of *TRY* and *TCL1*, two inhibitors of trichome development [240].

Recent studies across various plant species underscore the crucial role of auxins in orchestrating the coordination of seed life [241]. Building on this, a connection was established between the NAC family and auxins in the regulation of seed germination. Specifically, *NTM2* (*ANAC069*), identified as a plasmalemma-bound NAC TF, integrates both auxin and salt signals to regulate seed germination in Arabidopsis [242]. *NTM2* is induced by high salinity. Notably, the *ntm2-1* mutant, unresponsive to high salinity, exhibits resistance to salt-induced germination inhibition, suggesting a role for *NTM2* in the salt regulation of seed germination [242]. Intriguingly, *NTM2’*s function in salt signaling is independent of ABA and GAs. However, auxin influences the germination of *ntm2-1* seeds under salinity. Simultaneously, *NTM2* modulates auxin signaling through the INDOLE-3-ACETIC ACID 30 (*IAA30*) gene, linking salt signals to auxin signaling. In conditions of high salinity, auxins and the induction of the *IAA30* gene, seen in the WT but not in the *ntm2-1* mutant, contribute to delayed germination. In conclusion, *NTM2* negatively regulates seed germination under salt stress by upregulating *IAA30* expression [242]. Additionally, the NTL8 protein undergoes proteolytic activation in the presence of high salt, leading to the repression of the GA3 oxidase 1 (*GA3ox1*) gene in an ABA-independent manner [239]. Within the intricate network involving auxin, GAs, and salt signaling, a lingering question remains: how does the positive regulatory role of auxin on radicle emergence become compromised under high salinity conditions [242,243]?

## 7. Molecular Evolutionary Analysis of NAC TFs: A Derived Origen from WRKY

The transition to sessile life and terrestrial colonization of plants has been accompanied by significant genetic, developmental, and biochemical innovations in plant physiology. Consequently, the terrestrial colonization event led to a notable divergence and subsequent diversification of lineages adapted to either aquatic or terrestrial environments. One such example is the WRKY TF family, a group of proteins now recognized as crucial regulators of plant growth, development, and response to environmental stresses. The first WRKY protein, SWEET POTATO FACTOR1 (IbSPF1), was discovered in *Ipomoea batatas* [244]. A single copy of the WRKY gene, encoding two WRKY domains, was identified in the unicellular protist *Giardia lamblia*, a primitive unicellular eukaryote that diverged approximately ~1500 million years ago, as well as in the green alga *Chlamydomonas reinhardtii* [245]. It appears that the ancestral WRKY gene underwent numerous duplications throughout the evolution of plants, leading to the emergence of a large gene family in angiosperms [246]. Indeed, evolutionary analyses suggest that the C-terminal domain of the two-WRKY-domain encoding gene acts as the precursor to the single-WRKY-domain encoding genes. This observation indicates a possible evolutionary pathway where gene duplication events lead to the formation of new genes with modified structures and functions [247]. In the kingdom Plantae (Archaeplastida), the presence of WRKY proteins has been identified across the entire range of lineages studied from Rhodophyta to angiosperms [245]. These findings underscore the evolutionary significance and complex organization of WRKY proteins [248].

Currently, the evolutionary relationship between WRKY and NAC TFs is a topic of broad debate, mainly because no phylogenetic analysis encompassing both families has yet been achieved. The phylogenetic tree from this update (Figure 3) shows that members of both TF families are present in Charales, Zygnematales, and liverworts. However, Chlorophytes, including the model algae *Chlamidomonas reihardii*, which diverged earlier than the three aforementioned families, possess members of WRKY but not NAC. This observation supports the view that the WRKY family is evolutionarily older than NAC. Nevertheless, it is worth considering the possibility that the NAC family may have disappeared in Chlorophytes. Interestingly, the W-box elements for NAC and WRKY TFs exhibit high homology. Yet, some researchers do not support the idea of a close phylogenetic relationship between WRKY and NAC. Whereas the WRKY family is found in plants, fungi, and protists, suggesting an ancient evolutionary origin, the NAC family is present in plants and some algae, with its origin likely occurring later in evolutionary history. NAC transcription factors appeared before the emergence of land plants [24]. Evolutionary analysis indicated that segmental duplication and tandem duplication were the main mechanisms contributing to the expansion of the NAC gene family [249]. Structurally speaking, both WRKY and NAC proteins, along with the three more TFs families, belong to the GCM domain factors class [7]. WRKY and NAC share a common feature: a core β-sheet structure, with the edge of this β-sheet playing a crucial role in DNA recognition in a sequence-specific manner. This edge is a main part of the “WRKYGQK” motifs in WRKY and “WKATGTD” motifs in NAC. However, there are differences between WRKY and NAC: (i) WRKY possessed a Zinc-binding motif (not found in NAC); (ii) NAC proteins often have a dimerization arm, a structural motif not typically found in WRKY; (iii) WRKY often has two tandem repeated WRKY domains, whereas only a limited number of NAC exhibit such domain repetition; and (iv) in terms of DNA recognition, WRKY primarily utilizes one edge of the β-sheet (characterized by the ‘‘WRKYGQK” motifs) to interact with the major groove of DNA. In contrast, NAC employs an additional loop, positioned before the “WKATGTD” motifs, to recognize the minor groove of DNA as well [250]. To summarize, there are significant differences between NAC and WRKY, suggesting that they may have evolved from a common ancestor but subsequently diverged into two major TF groups in plants, which underscores the crucial role of NACs in plant evolution. Our tree (Figure 4) supports the view that NAC lineages diverged from WRKY lineages after the divergence of WRKY groups 1N and 1C (sensu Zhang and Wang, 2005 [247]) and prior to the emergence of WRKY genes encompassed in groups 2a; 2b, 2c, 2d, 2e and 3. Although certain WRKY and NAC may exhibit similarities in non-conserved regions apart from the structural domains, some authors reject the view of a direct relationship between WRKY and NAC based solely on these sequences (Bohan Liu, personal communication, 2024). In their view, similarities could be attributed to genetic recombination and sequence exchange processes during the course of evolutionary events.

## 8. New Findings (2020–2024), Concluding Remarks, and Challenges for the Future

A significant number of plant-specific NAC (NAM-ATAF-CUC) genes/proteins have been isolated and studied across various species, yielding several notable findings. Through research outcomes spanning the past two decades, it has become widely recognized that this TF family plays multiple roles in signaling regulatory networks. However, functional studies of *NAC* genes are still limited and only a few members of this family display well-known functions. Remarkably, particularly notable NAC TFs, including *GmNAC30*, *GmNAC81*, *ORE1* (*ANAC92*), *OsNAC2*, *NAP* (NAC-like and activated by AP3/PI), *ATAF1* (*ANAC002*), and *JUB1* (*ANAC042*), among others, have undergone thorough analysis at the whole-genome level, including phylogenetic analysis, collinearity analysis, gene structure, motif analysis, cis-element analysis, etc., and examination of gene expression patterns [254,255]. Through these analyses, they have been identified as central hubs regulating both plant development and stress responses. Let us take a look at some very recent significant studies that have not been previously discussed in detail. In soybean, *GmNAC30* and *GmNAC81* collaborate to regulate the expression of VPE. Likewise, *GmNAC81* may negatively affect drought tolerance, not only by activating VPE but also by suppressing ABA signaling [113]. These two findings suggest that *GmNAC81* acts as a converging link connecting signals of stress-induced cell death with drought responses [256]. In addition, asparagine(N)-rich protein (NRP) induces GmNAC030 and GmNAC081, which form a nuclear heterodimer to fully activate the expression of VPE, the executor of the cell death program via vacuole collapse. That is, the NRP-NAC-VPE cell death signaling conserved module integrates osmotic and ER stress into a signaling cascade that leads to a cell death fate [257]. To summarize, the integration of stress signals into a cell death pathway seems clear. On the other hand, the in vivo binding of ORE1 (ANAC092/ORESARA 1) to the promoters of two senescence-associated genes like *SAG29/SWEET15*, involved in nutrient re-mobilization and chlorophyll degradation, and *SINA1*, a RING E3 ligase, was demonstrated in Arabidopsis. That is, ORE1 directly regulates the expression of *SAG29* and *SINA1*. This feature supports the idea that ORE1 is a master regulator in the process of LS, a biological pathway strictly controlled by gene-regulatory networks [116]. Unfortunately, the mediator subunit required for ORE1 activity, as well as the mechanistic basis for the recognition of the sequence of related genes, remains uncertain. Nevertheless, it was recently demonstrated that MED19a and ORE1 physically interact. This interaction has been observed both in vitro and in vivo during the deficiency of nitrogen, an inducer of LS [258]. Overall, this event highlights an important discovery in plant biology regarding the regulation of LS and the molecular interactions involved in this process. Additionally, it has recently been demonstrated that ORE1 serves as an in vivo phosphorylation substrate for CPK1, a calcium-dependent protein kinase that regulates cell death [259]. In conclusion, elucidating the impact of CPK1-mediated phosphorylation on ORE1 function holds the potential to unveil novel regulatory mechanisms underlying cell fate decisions in response to environmental cues and stress stimuli. Recently, the crystal structure of the ORE1-NAC domain alone and its DNA-binding form were reported. The structure of DNA-bound ORE1-NAC revealed the molecular basis for nucleobase recognition and phosphate backbone interactions [55]. This information is crucial for understanding the precise mechanisms by which ORE1 regulates gene expression and constructs cellular responses to various stimuli.

Rice, as a well-studied species, has its orthologous counterpart to *ORE1* in Arabidopsis, known as *OsNAC2*, and shares similarities in function, particularly in regulating LS. Recent studies have shown that *OsNAC2* inhibits SA signaling [260]. It forms a stable nuclear complex with OsEREBP1, disrupting the transmission of SA-mediated signaling. This OsEREBP1-OsNAC2 complex plays a role in co-regulating the expression of target genes. Conversely, OsEREBP1, belonging to the AP2/ERF family, exerts positive regulatory effects on resistance to bacterial blight in rice. It achieves this by interacting with the disease resistance-related protein OsXb22a in the cytoplasm, where it stabilizes OsXb22a. These findings suggest that OsNAC2 plays a role in balancing the distribution of OsEREBP1 between the nucleus and cytoplasm. Additionally, *OsNAC2* may indirectly inhibit SA synthesis by regulating *OsICS1* (*Isochorismate synthase 1*), the SA receptor gene *OsNPR1* (*Nonexpressor of pathogenesis-related genes 1*), and the SA synthesis genes *OsPAL6* and *OsPAL3*, thereby modulating plant immunity. *OsNAC2* bound to the promotor of *OsICS1* and *OsNPR1* and suppressed their promotor activities [260]. In conclusion, Zhong et al. (2023) discovered that *OsNAC2* is a negative regulator in the resistance to a bacterial blight disease in rice. On the other hand, *OsNAC2* plays a crucial role in plant development by directly regulating key genes in the GAs pathway and positively modulating CK signaling. Additionally, OsNAC2 inhibits cell division and serves as an integrator of auxin, GAs, and CK pathways, thereby contributing to the regulation of plant height and root development [126]. This year, the characterization of *OsNAC103* revealed its role in repressing cell cycle progression, with OsCYCP2;1, a P-type cyclin protein, identified as a potential target gene [261]. Interestingly, these findings further contribute to our understanding of the regulatory networks governing cell cycle dynamics and plant growth mediated by NAC TFs in rice.

The *NAP* (NAC-like, activated by AP3/PI) genes found in plants are indeed important members of the NAC family, playing crucial roles in various biological processes, including SL [33,202]. NAPs are highly regulated by ABA. For instance, *AtNAP*, a senescence-specific TF that plays a key role in promoting LS, participates in GAs-mediated chlorophyll degradation by interacting with *GA-insensitive* (*GAI*) and *repressor of ga1–3* (*RGA*). The interaction subsequently impaired the transcriptional activities of NAP to induce the expression of *AAO3* and senescence-associated gene *SAG113* [262,263]. Recently, the AtNAP-*AtCKX3* (*cytokinin oxidase 3*) regulatory module was characterized [264]. The Y1H experiments demonstrated that *AtNAP* physically binds to the *cis*-element of the *AtCKX3* promoter to direct its expression. The expression of *AtCKX3*, regulated by *AtNAP*, is specific to the LS process, thereby modulating it. This study highlights the importance of this module in regulating CK levels and influencing the LS process [264]. This research further suggests that the AtNAP-AtCKX3 module is implicated in LS by bridging the interaction between two antagonistic plant hormones, ABA and CK. In 2023, a noteworthy breakthrough took place [265]. The authors showed that the regulatory module ABA-AtNAP-SAG113 PP2C controls leaf longevity by dephosphorylating SAG114 SnRK3.25 kinase, a direct target of the regulatory module. The senescence-associated gene SAG114, epistatic to SAG113 PP2C, encodes SnRK3.25 (Calcineurin B-like protein), a serine/threonine protein kinase specifically expressed during LS. Interestingly, SAG114 is localized in the Golgi apparatus. The phosphorylation/dephosphorylation process is yet to be studied [266]. In 2022, the CCCH-type zinc finger gene known as Strong Staygreen (*PvSSG*) was characterized in *Panicum virgatum. PvSSG*, through protein–protein interaction, repressed the DNA-binding efficiency of *PvNAPs*, thereby functioning as an impediment in the progression of LS. Interestingly, the Y2H, pull-down, BiFC, and co-immunoprecipitation (co-IP) analyses provided evidence that PvSSG (SL repressor factor) directly interacted with PvNAPs (SL accelerator factor) in the nucleus, suppressing PvNAPs-induced precocious LS [267]. This study reports a novel regulatory module in LS consisting of PvSSG and a pair of senescence-promoting TFs, PvNAPs. Therefore, the presence of the PvSSG–PvNAPs module demonstrates a finely coordinated process of chlorophyll catabolism and LS. These findings shed light on the regulatory mechanisms involved in LS in switchgrass and offer insight into how protein interactions can influence the progression of this and other similar processes.

As demonstrated throughout this review, the LS process is of singular importance in the life of plants and therefore is extensively studied (i.e., *OSNAC5*). In the current year, the Borrill group achieved a significant breakthrough in the NACs-LS relationship in wheat (*Triticum aestivum* L.) [268]. Thus, *TaNAC5-1*, an ortholog of *OSNAC5* expressed in senescing flag leaves, acts as a positive regulator of LS. To characterize *TaNAC5-1*, the authors employed missense mutations in *TaNAC5-A1* and *TaNAC5-B1* from a TILLING mutant population and also overexpressed *TaNAC5-A1* in wheat. Mutation in *TaNAC5-1* resulted in a delayed onset of flag LS, while overexpression of *TaNAC5-A1* led to a slightly earlier onset of senescence. Unfortunately, the DAP-seq analysis for *TaNAC5-1* yielded limited information. Nonetheless, two serine peptidases and an α/β gliadin gene were identified as potential targets of *TaNAC5-1* [268].

The importance of the ATAF-like NAC group in a diverse array of stress-related signaling processes (e.g., pathogenesis) is undeniable. This group consists of four genes in Arabidopsis and two in rice. *ATAF1* (*ANAC002*) and *ATAF2* (*ANAC081*) from Arabidopsis were among the first described NAC members. Analysis of the four Arabidopsis ATAF NAC promoter regions reveals an over-representation of both an ABA-responsive element (YACGTGGC) and G-box (CACGTG), implicating regulation by bZIP TFs and providing some plausible explanation of their ABA and stress responsiveness. Interestingly, *ATAF1* has emerged as a potential regulator of Brassinosteroids (BRs) catabolism, whereas *ATAF2* promotes LS [88]. Likewise, ATAF1,2, ANAC102, and CCA1 proteins exhibit both self and mutual physical interactions. Additionally, the expression of *ATAF2*, *ANAC102*, and *ATAF1* is suppressed by BRs [269]. *ANAC102* is involved in BRs catabolism and exhibits circadian regulation controlled by phytochromes, and its cellular localization remains uncertain [136,251,252,253,254,269,270,271,272].

The multifunctional *JUB1* (*ANAC042*) TF, whose expression is induced by H_2_O_2_, is involved in camalexin biosynthesis, which is the major phytoalexin in Arabidopsis. *JUB1* is also involved in the induction and forms the core of a robust regulatory module that triggers DELLA accumulation. Thus, the HB40-JUB1 regulatory network plays an important role in controlling GAs homeostasis during plant growth. HB40 directly activates the transcription of JUNGBRUNNEN1 (JUB1), a key TF that represses growth by suppressing GAs biosynthesis and signaling [271]. Previously, it was shown in tomato that *SlDREB1*, *SlDREB2*, and *SlDELLA* are potential target genes of SlJUB1 under drought stress. To summarize, *AtJUB1* directly inhibits the expression of GAs and BRs biosynthetic genes, resulting in the accumulation of DELLA protein. This accumulation inhibits growth and increases plant resistance to stresses. Recently, *GhJUB1L1* (homolog gene of *AtJUB1*) was proved to directly bind to the promoters of *GhABI1*, *GhSOS2*, *GhCCoAOMT1*, *GhCesA7*, and *GhIRX14* in vivo, resulting in the induction of the expression of these five genes in cotton [254]. Zhang et al. (2022) indicate that (i) the expression of *JUB1* was activated by CLE14 (CLAVATA3/ENDOSPERM SURROUNDING REGION); (ii) *JUB1* functions downstream of the *CLE14* signal; (iii) *CLE14* expression was also significantly induced by ABA, JA, and SA; (iv) CLE14 is involved in the regulation of ROS accumulation during senescence and under stress conditions, potentially by promoting ROS scavenging activities in a JUB1-dependent manner; and (v) the CLE14-JUB1-ROS module serves as a braking mechanism, in the form of a negative feedback loop, to prevent precocious cell death and ensure that the LS process is undertaken in an orderly manner, such that nutrient remobilization can be accomplished in time [272]. Taken together, the results from Zhang et al. (2022) show that the small peptide CLE14 functions as a novel ‘brake signal’ to regulate age-dependent and stress-induced LS through JUB1-mediated ROS scavenging.

Very recently, and to conclude this section, two adjacent *NAC* genes named *PpNAC1* and *PpNAC5* were functionally characterized in peach (*Prunus persica* L.), demonstrating their roles in regulating fruit maturity date and flavor [136]. The Han group, demonstrated that *PpNAC1* and *PpNAC5*, physically separated by a 19 863-bp DNA region, had several effects as follows: (i) They had pleiotropic effects on plant development, increase in organic acid metabolism and sugar accumulation by activating sugar transport genes. (ii) They participated in organic acid accumulation by activating the expression of the *PpGAD* gene, which encodes glutamate decarboxylase, contributing to citrate degradation and acidity reduction in peach. Therefore, the *GAD* gene appears to be a promising target for genetic improvement aimed at enhancing citrate levels in fruits. (iii) They interacted with each other in vivo (Y1H) to form heterodimer complexes that are conserved in eudicots. (iv) Both TFs functioned as ripening enhancers, with the former having a stronger ripening acceleration effect (divergence between their C-terminal regions?), and *PpNAC5* could be more crucial for fruit enlargement than for fruit ripening. (v) The orthologs of both TFs were found in all tested angiosperms, but not in gymnosperms [136]. Finally, the results obtained to date demonstrate that *NAC1* or *NAC5* orthologues are not functionally exchangeable in dry fruit-bearing angiosperms.

The methodologies used for studying the NAC family of TFs have remained relatively consistent over the last twenty years, with a focus on whole-genome analysis and gene expression studies. Techniques such as next-generation sequencing, microarray analysis, and quantitative PCR have been instrumental in elucidating the functions and regulatory roles of NAC genes. Despite the methodological continuity, advances in bioinformatics tools, data analysis algorithms, and high-throughput sequencing technologies have enhanced the efficiency and depth of our understanding of the NAC family. Additionally, the integration of multi-omics approaches, such as combining genomics, transcriptomics, proteomics, and metabolomics data, has provided comprehensive insights into the regulatory networks and functional dynamics of NAC TFs. While the core methodologies may have remained consistent, the refinement and integration of these techniques have undoubtedly contributed to significant advancements in our understanding of the NAC family and its roles in plant development, stress responses, and other biological processes. However, the study of the interaction between TFs and chromatin in vivo is an aspect that is still to be addressed. So far, TFs have been studied in isolation. However, in vivo, TFs often form complexes with other TFs or with other factors (co-repressors, regulators, etc.) (see Section 5). The study of complexes in vivo is lacking; that is, it is necessary to address the study of the complex TF-DNA interaction in vivo. The state of chromatin is crucial for TF function. It is necessary to combine chromatin analysis (e.g., ATAC-seq) with the study of TF binding sites (preferably complexes of TFs-regulatory proteins) (e.g., ChIP-seq). For the study of complexes, techniques such as TAP-tagging or Turbo-ID should be used and combined with studies of Hi-C or similar methods for the 3D chromatin structure. These studies should ideally be dynamic to observe how these complexes vary over time or in response to stress.

Indeed, it would be very interesting to study the mechanisms and endogenous and/or exogenous signals operating in the cell to distribute the NAC TFs in tethering and non-tethering configurations. The phylogenetic investigation into the origin of the NAC family in relation to the WRKY family is also of significance. Figure 4 depicts an accurate and simplified phylogeny of the major lineages in Viridiplantae. In this phylogeny, Charales (specifically *Chara braunii*) is the first lineage in which the NAC TF appears. 

## Figures and Tables

**Figure 1 ijms-25-05369-f001:**
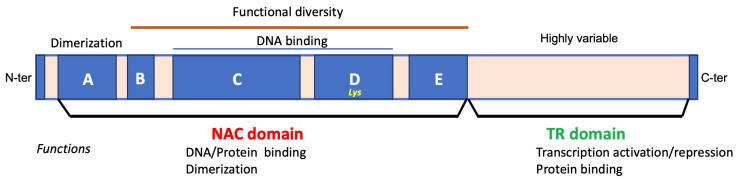
TFs exhibit characteristic domains responsible for various functions, including DNA binding, oligomerization/protein–protein interactions, transcriptional regulation, and nuclear localization. The accompanying figure illustrates the structure of a NAC-TF. The NAC domain, enclosed by a red ellipse, comprises nearly 150 amino acid residues and often includes a nuclear localization signal, enabling protein binding. Subdomain A facilitates protein dimerization, subdomains B and E contribute to functional diversity, while subdomains C and D, which are positively charged and highly conserved, are responsible for DNA binding. The C-terminal transcriptional regulatory (TR) region, surrounded by a green ellipse, functions as a transcriptional activator or repressor and may possess protein binding activity, interacting with other TFs. Adapted from Singh [33] and Diao [57].

**Figure 2 ijms-25-05369-f002:**
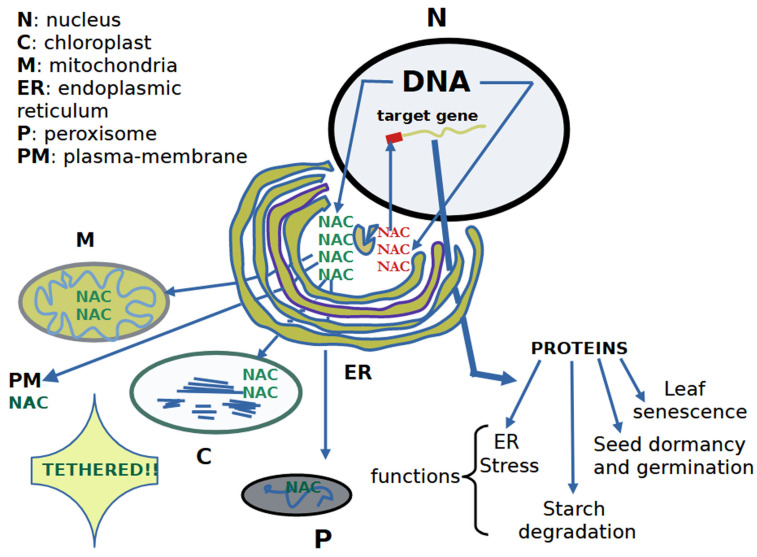
The cellular synthesis of NACs generates a population of these TFs in the ER. The nuclear genes responsible are affected by phytohormones. A notable portion of the NAC population is sent to the nucleus to alter the transcription of target genes, producing the corresponding proteins. These proteins alter a series of physiological processes such as LS, seed reserve degradation (e.g., starch), ER stress, and seed dormancy and germination, among others. The remaining endoplasmic population of NACs (membrane-tethered NAC TFs) is sent to cellular compartments (i.e., chloroplasts, peroxisomes, and mitochondria) and plasma membranes. The membrane-tethered subset is a small family specific to plants, which lose their transmembrane domain and are then sent to the nucleus to exert their physiological role once bound to the corresponding target genes. The exit of membrane-tethered NAC TF from the corresponding cellular compartment occurs in response to environmental and developmental changes. For further information, see [71,76,77].

**Figure 3 ijms-25-05369-f003:**
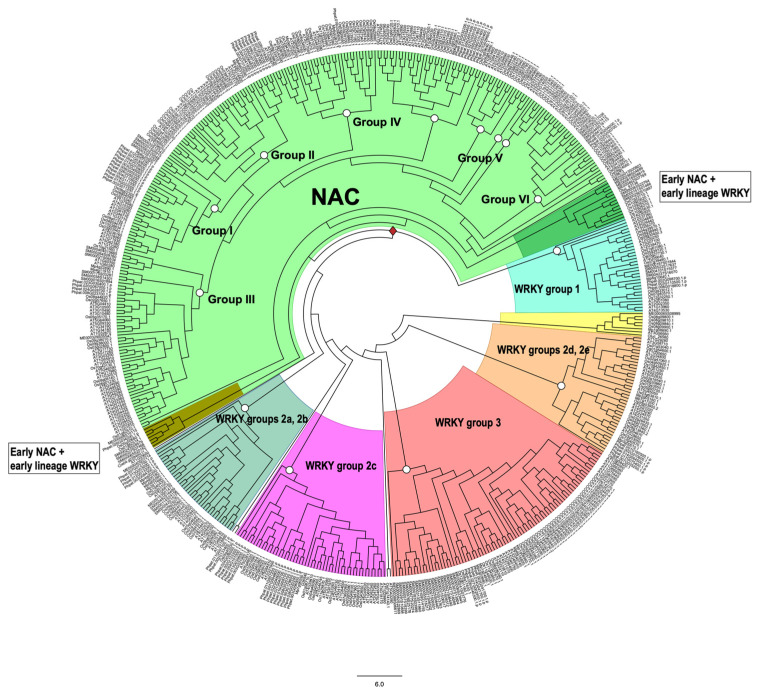
Maximum-Likelihood phylogenetic analysis of 266 NAC and 268 WRKY protein sequences, aligned with ClustalW, performed in IQ-TREE (2021) under a JTT + R7 model chosen according to BIC. Red diamond indicates the node (BS 85%) showing a sister relationship between WRKY group 1 and NAC genes. White circles designate main groups of NAC genes following Pereira Santana et al. (2015) [251], and WRKY genes according to Zhang and Wang (2005) [247].

**Figure 4 ijms-25-05369-f004:**
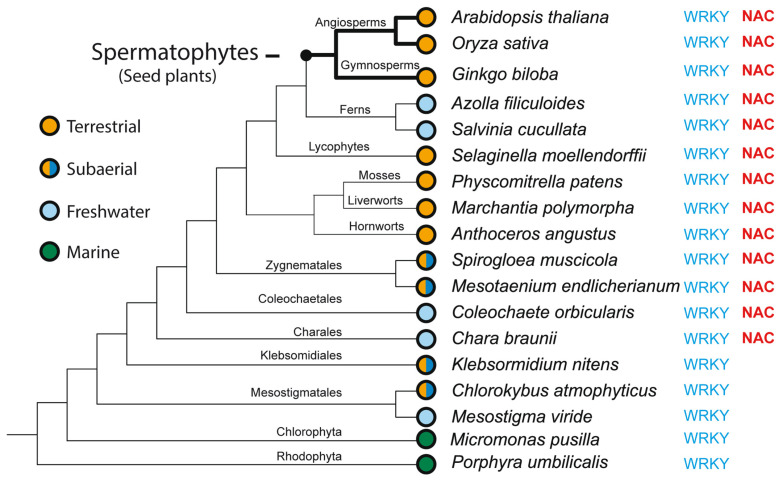
This tree indicates that Klebsomidiales is the last lineage without NAC TFs, while Charales is the first lineage with WRKY and NACs. For each species, the habitat and documented occurrence of WRKY and NAC gene families are depicted. Adapted from Leebens-Mack et al. 2019 [252] and Wang et al. 2021 [253].

**Table 1 ijms-25-05369-t001:** Selected species in which NAC TFs have been compiled in this study.

Species	References
*Arabidopsis thaliana*	[151,169,176]
*Capsicum annuum*	[159]
*Citrullus lanatus*	[177,178,179]
*Cucumis melo*	[160]
*Dendrobium nobile*	[78]
*Fagopyrum tataricum*	[68]
*Glycine max*	[31,40,48,113,115,121,154,180]
*Helianthus annus*	[46,162,181]
*Klebsormidium flaccidum*	[75]
*Liriodendron tulipifera*	[4]
*Malus domestica*	[161]
*Manihot sculenta*	[157]
*Medicago truncatula*	[173]
*Nicotiana tabacum*	[172]
*Oryza sativa*	[42,52,54,95,125,127,148,152,176,182,183,184]
*Passiflora edulis*	[69]
*Pinus tabuliformis*	[60]
*Populus trichocarpa*	[170]
*Saccharum spontaneum*	[163]
*Solanum lycopersicum*	[34,53,67,134]
*Solanum tuberosum*	[61,156]
*Triticum aestivum*	[43,46,147,153,185,186,187]
*Vitis vinifera*	[171]
*Zea mays*	[44,138,139,140,141,144,145,146,155,188,189,190,191]
*Brassicaceae*	[71,158,166,174]
*Other species*	[164,165,167,168,175]

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
