# Peer review of "Transcriptional Control of Seed Life: New Insights into the Role of the NAC Family"

_ijms, 2024, doi:10.3390/ijms25105369_

Round 1
Reviewer 1 Report
Comments and Suggestions for Authors
General comments:
The idea of the review is interesting, but the focus on plant science is more evident than on seed science. My suggestion is to change the perspective for the paper to be accepted. At the present way, the paper needs to be improved for seed science.
The title is inappropriate for the text. Just a small part of the text is concerned with seed biology. The better title should be ‘New Insights into Role for NAC Family on Plant Science’.
The abstract is appropriate for the issues focused on the paper.
Minor suggestions in the abstract –
change dicots for eudicots (please, throughout the text)
In keywords, cut off ‘terrestrial colonization’. In addition, to reduce the number of keywords to 5 or 6. A suggestion is to cut off NAC modules, leaf 38 senescence, NAC modules and phylogenetics.
In the Introduction section, the authors provide a gentle introduction to TFs and NAC. A reader-friendly text. However, it is needed for a hypothesis and/or an aim to be shown clearly to reach a global idea and give up security to the reader for following the main text.
Other subtitles throughout the text are coherent, but the central idea is not about seed science. The authors have revised the literature concerned with NACs for plant science. That must be considered.
Author Response
see two text

Reviewer 2 Report
Comments and Suggestions for Authors
The review provides an update on the NAC transcription factors (TFs) family in plant development, physiology, and abiotic/biotic stress responses. The manuscript provides am extensive and detailed literature review of the NAC TFs functions and implications in various plant pathways supported by numerous references. The authors have performed an enormous amount of work to collect, organize and present this wealth of information in a multi-page review manuscript. However, the manuscript lacks a concise and meticulous focus as well as coherence to the main topic investigated, that is the seed. Furthermore, due to its length and the lack of tables and figures, except two, illustrating vital functions/effects of NAC TFs in seed life, it seems rather impossible to fully read, comprehend and/or engage readership. Therefore, the manuscript should be re-organized and structured to provide a well-focused and timely update of NAC TFs in seed life. Some indicative issues that the authors are encouraged to elaborate and clarify areas follows:
1. If the seed is the plant organ chosen to be explored in this literature review, then it should be meticulously reviewed providing information and knowledge gained from research studies on the NAC TFs implications in the seed developmental stages and parts (i.e. formation, development, filling, maturation, imbibition, germination, embryo, endosperm, seed coat, etc.). These can be also illustrated with figures depicting the specific involvements of NAC TFs at various steps of the processes.
2. Similarly, the introduction of Tables presenting cumulative information of the number of NAC TFs identified/ found to be implicated in the above processes and characterized for their action(s) along with the plant species explored could provide a cumulative view of the research completed to date.
3. As the amount of information could be overwhelming it is suggested to center the review on angiosperms plants, and/or a particular group of them (i.e. annuals, perennials, etc.) providing a concise overview of the most important research on the topic.
4. A clear description of abbreviations used should be shown upon first note.
5. Sections that provide general background on TFs could be omitted such as Ln 44-83. While others should be thoroughly elaborated and edited to be concise to the topic explored, are as follows: ln 261-560. Undoubtedly plant hormones have a distinct interaction with TFs; however this should be reviewed in light of the specific topic explored, the NAC TFs in seed life.
6. Moreover, the ln 1227-1295, section 7, regarding the molecular evolutionary analysis of NAC TFs seems rather irrelevant, as the review does not dissect the evolutionary interactions of plant TFs. Thus, suggestion is to be omitted along with Figure 2.
7. Similarly, sections that explore the interactions of NAC TFs with other transcription factors such as WRKY should be thoroughly edited as it is known NAC TFs are involved in numerous plant pathways along with bHLH, MYB, WRKY, etc.
8. The final section titled “Concluding Remarks and Challenges for the Future” should be more concise to the topic explored, highlighting the conclusions and future challenges and research perspectives.
In conclusion, the review should be re-organized and structured to center closely on the explored topic, providing a cohesive overview of current research advancements and future prospects.
Comments on the Quality of English LanguageMinor english editing .
Author Response
see two text

Round 2
Reviewer 1 Report
Comments and Suggestions for Authors
All issues were addressed. Therefore, I suggest publishing the paper as it is presented.
Author Response
Thank you for your review.
Reviewer 2 Report
Comments and Suggestions for Authors
In the revised version of the manuscript the authors have addressed partly the reviewer’s comments. Although the change in the title provides a more general aspect of the review topic, the organization and structure of the manuscript did not change, resulting in lack of a clear and close connection/correlation to the objective of the Journal’s Special Issue.
Undoubtedly the wealth of information presented in the manuscript regarding the varied roles of the NAC transcription factors (TFs) in plant’s processes is overwhelming. However, recent reviews are available with schematic representations of the NAC genes roles and functions in plant development, regulation of abiotic/biotic stress tolerance, secondary cell wall synthesis, lateral root development, ROS signaling, leaf senescence, programmed cell death, crosstalk with phytohormones, etc. The authors failed to provide a more focused review regarding the role of the NAC genes on the seed, the central unit of the plant life, as previously suggested. The picture of the cellular and molecular mechanisms of NAC involvement during seed life is not clearly depicted, to expand our understanding of the fundamental mechanisms to stimulate transformation in agriculture and crop improvement strategies to address the urgent food security challenges in the climate crisis times.
Therefore, I regret to consider that in its present state the revised manuscript does not fulfill the objective of the SI, nor are the responses to the comments adequately documented.
Comments on the Quality of English LanguageMinor editing of English is required.
